# Calaxin stabilizes the docking of outer arm dyneins onto ciliary doublet microtubule in vertebrates

Hiroshi Yamaguchi, Motohiro Morikawa, Masahide Kikkawa*

Department of Cell Biology and Anatomy, Graduate School of Medicine, The University of Tokyo, Tokyo, Japan

**Abstract** Outer arm dynein (OAD) is the main force generator of ciliary beating. Although OAD loss is the most frequent cause of human primary ciliary dyskinesia, the docking mechanism of OAD onto the ciliary doublet microtubule (DMT) remains elusive in vertebrates. Here, we analyzed the functions of Calaxin/Efcab1 and Armc4, the two of five components of vertebrate OAD-DC (docking complex), using zebrafish spermatozoa and cryo-electron tomography. Mutation of *armc4* caused complete loss of OAD, whereas mutation of *calaxin* caused only partial loss of OAD. Detailed structural analysis revealed that *calaxin*$^{-/-}$ OADs are tethered to DMT through DC components other than Calaxin, and that recombinant Calaxin can autonomously rescue the deficient DC structure and the OAD instability. Our data demonstrate the discrete roles of Calaxin and Armc4 in the OAD-DMT interaction, suggesting the stabilizing process of OAD docking onto DMT in vertebrates.

## Editor's evaluation

In vertebrates, ciliary motility is powered by axonemal dyneins, known as OADs, tethered to doublet microtubules by a pentameric docking complex including the Armc4 and Calaxin subunits. This valuable study combines zebrafish genetics with cryo-electron tomography to convincingly show that Armc4 plays a critical role in the docking of OAD and that Calaxin stabilizes the molecular interaction. The work will be of interest to those studying the structure and function of the axoneme, and motile cilia in general.

*For correspondence: mkikkawa@m.u-tokyo.ac.jp

Competing interest: The authors declare that no competing interests exist.

## Introduction

Motile cilia/flagella are evolutionarily conserved hair-like protrusions of eukaryotic cells. Ciliary motility is responsible for various biological processes, including locomotion of unicellular organisms, vertebrate left-right patterning, cerebrospinal fluid flow, airway mucociliary clearance, and swimming of spermatozoa. The ciliary cytoskeleton is called an axoneme, which consists of nine peripheral doublet microtubules (DMTs) with central-pair microtubules (so-called 9+2 structure) or without central-pair microtubules (so-called 9+0). The beating force of cilia/flagella is generated by axonemal dyneins, which make two rows of projections on the DMT: outer arm dynein (OAD) and inner arm dynein (IAD). The OAD row consists of single large dynein molecules that repeat every 24 nm. The IAD row consists of seven different types of dyneins (IAD a to g) that each repeat every 96 nm (*Oda et al., 2014*). Although each dynein type can generate the motive force of ciliary motility, OADs mainly influence the frequency and the power output of the beating, while IADs affect the waveform amplitude (*Brokaw, 1994*; *Brokaw and Kamiya, 1987*; *Kamiya, 2002*).

In humans, malfunctions of motile cilia often cause primary ciliary dyskinesia (PCD). PCD is a genetically heterogeneous, inherited disorder characterized by inversions of visceral laterality, hydrocephalus,

recurrent respiratory infections, and male infertility. OAD defects are the most common cause of PCD; mutations of over 40 genes are known to cause PCD, including 24 genes related to OAD malfunctions (*Wallmeier et al., 2020*; *Antony et al., 2021*). These variations of OAD-related PCD genes reflect the multiple processes of OAD construction described below. OAD is a large protein complex composed of two heavy chains (HCs), two intermediate chains, and numerous light chains. OAD subunits first undergo cytoplasmic pre-assembly before transport into cilia, which requires the support of multiple proteins called DNAAFs (dynein axonemal assembly factors) (*Kobayashi and Takeda, 2012*; *Mitchison et al., 2012*; *Yamaguchi et al., 2018*; *Braschi et al., 2022*). After assembly, OADs are targeted into the ciliary compartment, where the intraflagellar transport (IFT) machinery carries OADs along the DMT (*Kozminski et al., 1993*; *Klena and Pigino, 2022*). At the final position of OAD, docking complex (DC) proteins interact with OADs and array them on the DMT.

Although OAD structures are mostly conserved among eukaryotes, there are several differences reflecting variations in ciliary motilities. For example, OADs in green algae *Chlamydomonas* and ciliate *Tetrahymena* have three HCs (α-, β-, and γ-HC), while metazoan OADs have only two HCs (β- and γ-HC). DC structures also differ among organisms. *Chlamydomonas* DC is composed of three subunits: DC1, DC2, and DC3 (*Takada and Kamiya, 1994*; *Koutoulis et al., 1997*; *Wakabayashi et al., 2001*; *Takada et al., 2002*). The DC1/DC2 coiled-coil and the globular DC3 work as two linker structures tethering OAD to DMT (*Walton et al., 2021*). On the other hand, vertebrates have pentameric DC compositions: CCDC151, CCDC114, TTC25, ARMC4, and Calaxin (*Gui et al., 2021*). Based on the function of these proteins as components of outer dynein arm docking-complex (ODAD), they are also known as ODAD1 (CCDC114), ODAD2 (ARMC4), ODAD3 (CCDC151), ODAD4 (TTC25), and ODAD5 (Calaxin/CLXN). CCDC151 and CCDC114 are homologous to *Chlamydomonas* DC1 and DC2, respectively. Orthologues of TTC25, ARMC4, and Calaxin are not defined in *Chlamydomonas*, indicating the evolutionary divergence of DC functions. As OADs generate the main force of ciliary beating, proper construction of OAD-DMT is essential for motile cilia. However, the docking process of OAD and the functions of DC components remain elusive in vertebrates.

Calaxin was first identified as a calcium sensor protein of OAD in ascidian *Ciona intestinalis* (*Mizuno et al., 2009*). *Ciona* Calaxin plays important roles in sperm chemotaxis by modulating OAD activity depending on Ca²⁺ concentration (*Mizuno et al., 2012*; *Inaba, 2015*). To analyze the Calaxin function in vertebrates, our group previously generated knockout mutants in mice and zebrafish (*Sasaki et al., 2019*). *Calaxin* mutants showed abnormal ciliary motilities not only in sperm flagella, but also in tracheal cilia, brain ependymal cilia, and left-right organizer cilia. These phenotypes indicate that Calaxin is essential for proper ciliary motilities, which are not necessarily related to the Ca²⁺-dependent sperm chemotaxis. Interestingly, *Calaxin*⁻/⁻ OADs seemed mostly intact when observed by conventional transmission electron microscopy. Unlike Calaxin, other DC components (CCDC151, CCDC114, TTC25, and ARMC4) were already reported as human PCD-causative genes (*Hjeij et al., 2014*; *Onoufriadis et al., 2013*; *Knowles et al., 2013*; *Wallmeier et al., 2016*; *Hjeij et al., 2013*). Mutations of these genes cause complete or near-complete loss of OAD in the patient's respiratory cilia.

In this study, we further analyzed the function of Calaxin using zebrafish spermatozoa and cryo-electron tomography (cryo-ET). In addition to the *calaxin*⁻/⁻, we generated the zebrafish mutant of *armc4*, the largest component of vertebrate DC, to assess the function of other DC components. *armc4* mutation caused complete loss of OAD and slower sperm beating. In *calaxin*⁻/⁻, however, OAD was only partially lost at the distal region of sperm flagella, which was correlated with abnormal sperm waveform. Cryo-ET analysis revealed the detailed structure of OAD-DC in WT and *calaxin*⁻/⁻. The remaining OADs in *calaxin*⁻/⁻ were tethered to DMT through DC components other than Calaxin, suggesting that Calaxin works to stabilize the OAD-DMT interaction. Armc4 was required for the ciliary localization of Calaxin, which implied the ciliary targeting of DC components as pre-assembled complexes. We also report the slight conformation change of vertebrate DC at higher Ca²⁺ concentration, in line with the calcium sensor function of the Calaxin.

## Results

### *calaxin*⁻/⁻ and *armc4*⁻/⁻ zebrafish exhibit PCD phenotype of laterality defect

Previously, our group generated the *calaxin*⁻/⁻ zebrafish line and reported the phenotype of Kupffer's vesicle cilia (*Sasaki et al., 2019*; referred to as *efcab1*⁻/⁻ zebrafish). Kupffer's vesicle is the left-right organizer of teleost fish and orthologous to the mammalian embryonic node, in which rotating cilia produce leftward fluid flow to determine the left-right body axis (*Essner et al., 2005*; *Hirokawa et al., 2012*; *Shinohara and Hamada, 2017*). *calaxin* mutation caused slower rotation or irregular motion of Kupffer's vesicle cilia, with the consequent randomization of left-right patterning.

To compare the functions of different DC components, we also generated zebrafish *armc4* mutants. A stop codon was inserted into exon2 of the *armc4* gene by CRISPR/Cas9 (*Figure 1A-C*). Since *armc4*⁻/⁻ zebrafish were viable, we established a homozygous mutant line. We observed the Kupffer's vesicle cilia of *armc4*⁻/⁻ and found abnormal ciliary motilities: slower rotation, irregular motion, or immotility (*Figure 1D–F*; *Video 1*). These ciliary motility defects induced laterality randomization, as inverted heart looping was observed in almost half of the *armc4*⁻/⁻ embryos (*Figure 1G*). For comparison at a glance, we displayed *calaxin*⁻/⁻ data from our previous study (*Sasaki et al., 2019*) in *Figure 1E–G*. Although both *calaxin*⁻/⁻ and *armc4*⁻/⁻ showed laterality defects with abnormal ciliary motilities, immotile cilia were found only in *armc4*⁻/⁻ (*Figure 1E*, red), indicating the more severe impact of *armc4* mutation on the proper ciliary motilities.

### *calaxin*⁻/⁻ and *armc4*⁻/⁻ spermatozoa show OAD loss and abnormal motilities

We next focused on the sperm phenotypes and performed immunofluorescence microscopy. Dnah8, the OAD γ-HC component, was localized along the entire length of WT sperm flagella (*Figure 2*). In *calaxin*⁻/⁻, spermatozoa showed partial loss of Dnah8 at the distal region of the flagella (*Figure 2A*, white arrowheads). Dnah8 signal was detected in the proximal two-thirds of *calaxin*⁻/⁻ flagella, but the border of the signal was unclear, suggesting that OADs decreased gradually toward the distal end. *armc4*⁻/⁻ showed complete loss of Dnah8 (*Figure 2A*, asterisk). Calaxin signals were absent not only in *calaxin*⁻/⁻ but also in *armc4*⁻/⁻ (*Figure 2B*, asterisks). We also assessed the localization of IADs by the immunostaining of Dnah2, the IAD-f β-HC component. Dnah2 was distributed normally along the entire length of flagella in both mutants (*Figure 2C*).

To correlate the OAD defects with ciliary motilities, we analyzed the sperm waveforms using a high-speed camera. We tracked the heads of swimming spermatozoa and generated the waveform traces and the shear angle plots (*Figure 2D*; *Video 2*). Sperm flagella were traced at every 1 ms in WT and *calaxin*⁻/⁻, while 2 ms intervals were used in *armc4*⁻/⁻ to be adjusted for the decreased beating frequency. WT flagella exhibited a sine wave motion, with the shear angle plots showing constant bend propagation. *calaxin*⁻/⁻ flagella exhibited disturbed bend propagation at the distal region, as the shear angle plots lost their slopes (*Figure 2D*, black arrowhead). This distal-specific phenotype can be correlated with the loss of distal OAD in immunofluorescence microscopy. *armc4*⁻/⁻ flagella showed significantly slower beatings, consistent with the complete loss of OAD in immunofluorescence microscopy. Despite the OAD loss, *armc4*⁻/⁻ waveforms were mostly normal, suggesting that remaining IADs can sufficiently generate the sine wave motion of the sperm flagella.

We also tested the sperm motilities by CASA (computer-assisted sperm analysis) modified for zebrafish (*Wilson-Leedy and Ingermann, 2007*). In both mutants, the ratios of motile spermatozoa decreased significantly (*Figure 2E*). Swimming velocity (VAP) and beating frequency (BCF) were calculated from the trajectories of the motile spermatozoa, which have 20 µm/s or more velocities (*Figure 2F–G*; *Figure 2—figure supplement 2*; *Video 3*). Compared to WT, *calaxin*⁻/⁻ showed slightly increased swimming velocity but decreased beating frequency. In *armc4*⁻/⁻, both swimming velocity and beating frequency decreased significantly compared to WT and *calaxin*⁻/⁻. These data were consistent with the sperm motilities observed in the waveform analysis.

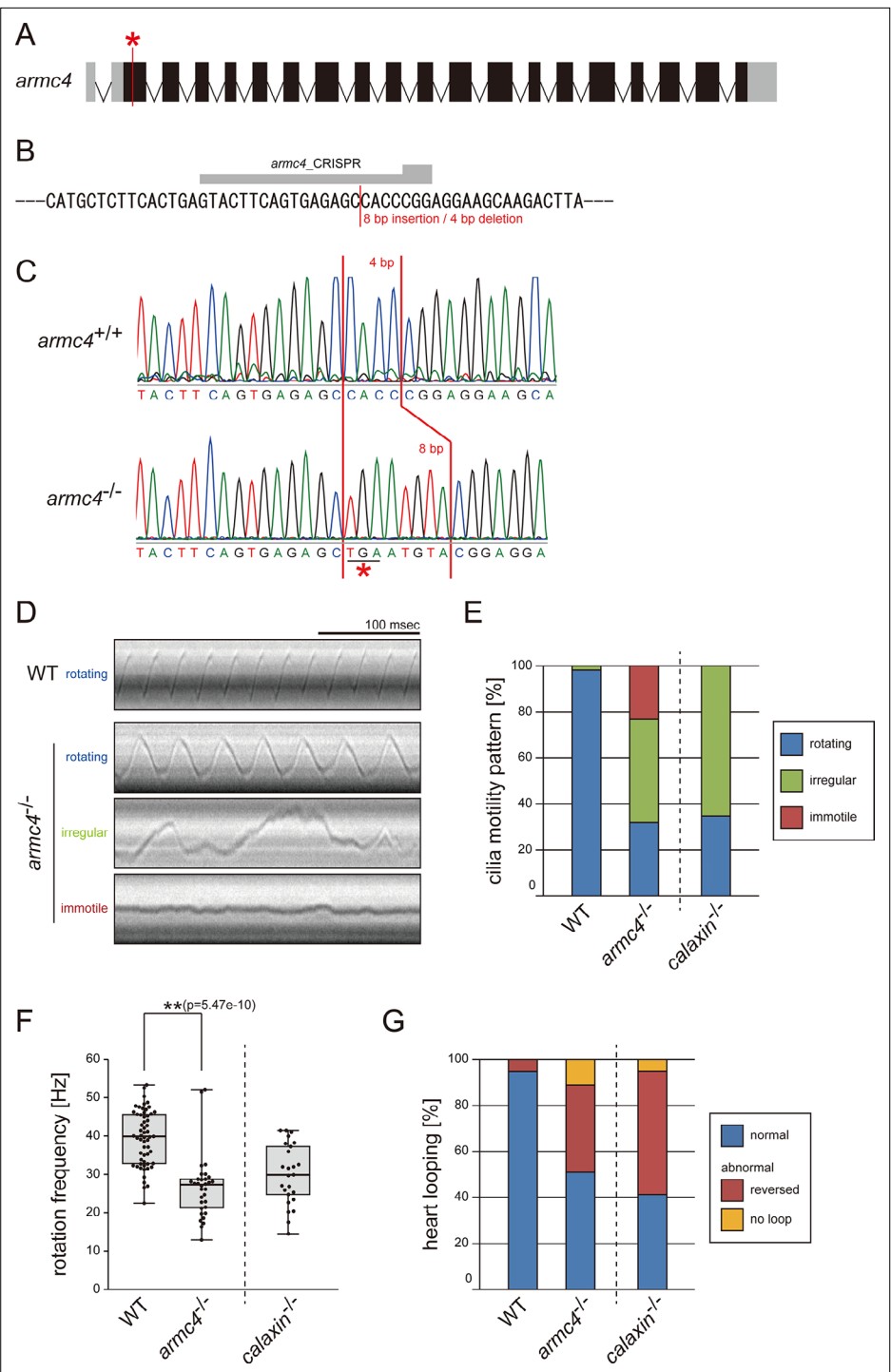

**Figure 1.** Mutation of *armc4* causes abnormal motility of Kupffer's vesicle cilia. (**A**) Genomic organization of zebrafish *armc4* gene. Black boxes: exons. Gray boxes: untranslated regions. Red asterisk: the genome-editing target site. (**B**) CRISPR/Cas9 target sequence. (**C**) Sanger sequencing of *armc4*⁺⁄⁺ and *armc4*⁻⁄⁻ fish around the genome-editing target site. The 8 bp-insertion in *armc4*⁻⁄⁻ includes a stop codon (red asterisk). (**D**) Typical kymographs of Kupffer's vesicle cilia in WT and *armc4*⁻⁄⁻ embryos. Kymograph patterns were categorized into three classes: rotating (blue), irregular (green), and immotile (red). Scale bar: 100 ms. (**E**) Ratios of each motility class. Number of cilia: 58 (WT) and 100 (*armc4*⁻⁄⁻). (**F**) Rotational frequencies of Kupffer's vesicle cilia. Number of cilia: 58 (WT) and 32 (*armc4*⁻⁄⁻). Boxes correspond to the first and third quartiles, lines inside the boxes indicate the medians, and whiskers extend to the full range of the data. p-Value was calculated with Welch's t-test.

*Figure 1 continued on next page*

*Figure 1 continued*

(**G**) Directions of heart looping. Number of embryos: 110 (WT) and 63 (*armc4*$^{-/-}$). For comparison, *calaxin*$^{-/-}$ data from *Sasaki et al., 2019* are displayed in (**E, F, and G**).

The online version of this article includes the following source data and figure supplement(s) for figure 1:

**Source data 1.** Numerical data of *Figure 1E*.

**Source data 2.** Numerical data of *Figure 1F*.

**Source data 3.** Numerical data of *Figure 1G*.

**Figure supplement 1.** Heart looping of mutant embryos at 36 hpf.

## Cryo-electron tomography reveals the ultrastructure of WT and mutant sperm DMTs

To gain structural insights into the mutant sperm phenotypes, we performed the cryo-ET analysis of sperm axonemes. Subtomographic averaging revealed the ultrastructure of the 96 nm repeat unit of DMT (*Figure 3A*; *Video 4*). This 96 nm repeat unit contains four OADs, seven IADs, three radial spokes (RSs), and a nexin-dynein regulatory complex (N-DRC; *Figure 3B*). Note that the local refinement was performed with four subdivided parts of the 96 nm repeat unit (DMT with axonemal dyneins, RS1, RS2, and RS3), because the RS-DMT connections were flexible. After local refinement, subdivided parts were combined to generate the whole structure of the 96 nm repeat unit (*Figure 3—figure supplement 1*).

Our improved cryo-ET analysis revealed the detailed structural features of zebrafish DMT. We previously reported the zebrafish DMT structure with a resolution of 42.5 Å (*Yamaguchi et al., 2018*). In this study, we increased the number of averaged particles approximately six-fold to get more reliable structures, with a resolution of ~22 Å (*Figure 3—figure supplement 2D*). For each axonemal dynein, the ring structure of the motor domain was observed (*Figure 3A*; *Video 4*). Each OAD has two motor domains, while each IAD has only one motor domain, except IAD-f, which has two motor domains with a large IC/LC (intermediate chain and light chain) (*Figure 3A–B*). As for RSs, the spoke heads of RS1 and RS2 showed similar structures and resembled a pair of skis (*Figure 3A*, black arrowheads). The RS3 spoke head showed a distorted star-like structure (*Figure 3A*, white arrowhead). Although the proximal part of RS3 appears structurally divergent among vertebrates (*Figure 3A and C*, red arrowheads), for RS heads and axonemal dyneins, similar features to zebrafish DMT were reported in sea urchin sperm flagella (*Lin et al., 2012*) and human respiratory cilia (*Lin et al., 2014*), suggesting that the DMT structures are mostly conserved among metazoans.

Mutant axonemes were also processed as described above to observe the DMT structures. In *calaxin*$^{-/-}$, some tomograms appeared to lose OADs in the axoneme, while the others retained OADs. To analyze these different axonemal features separately, we performed structural classification and sorted the subtomograms into OAD+ and OAD- classes. The OAD+ class showed a mostly normal OAD structure (*Figure 3D*; *Video 5*), while the OAD- class lost OAD (*Figure 3E*, red dotted circles; *Video 6*). The *armc4*$^{-/-}$ DMT showed complete loss of OAD (*Figure 3F*, red dotted circles; *Video 7*), consistent with the immunofluorescence microscopy data. Although OAD loss was found in both mutants, no significant defect was observed in other DMT components, such as IADs and RSs, indicating that the functions of Calaxin and Armc4 are OAD-specific.

## *calaxin*$^{-/-}$ OADs are attached to DMT through DC components other than Calaxin

To understand the cause of OAD loss, we further analyzed the OAD structures in WT and *calaxin*$^{-/-}$. For the local refinement of OAD structures, four OAD repeats were extracted from the 96 nm repeat unit of DMT. This fourfold increase in the particle number contributed to the improved

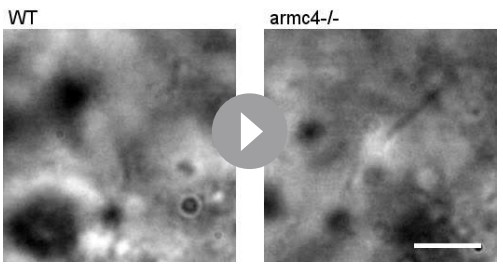

**Video 1.** Motilities of Kupffer's vesicle cilia in WT and *armc4*$^{-/-}$. Typical movies of Kupffer's vesicle cilia, filmed by a high-speed camera at 1000 fps and played at 30 fps. Scale bar: 5 μm.

https://elifesciences.org/articles/84860/figures#video1

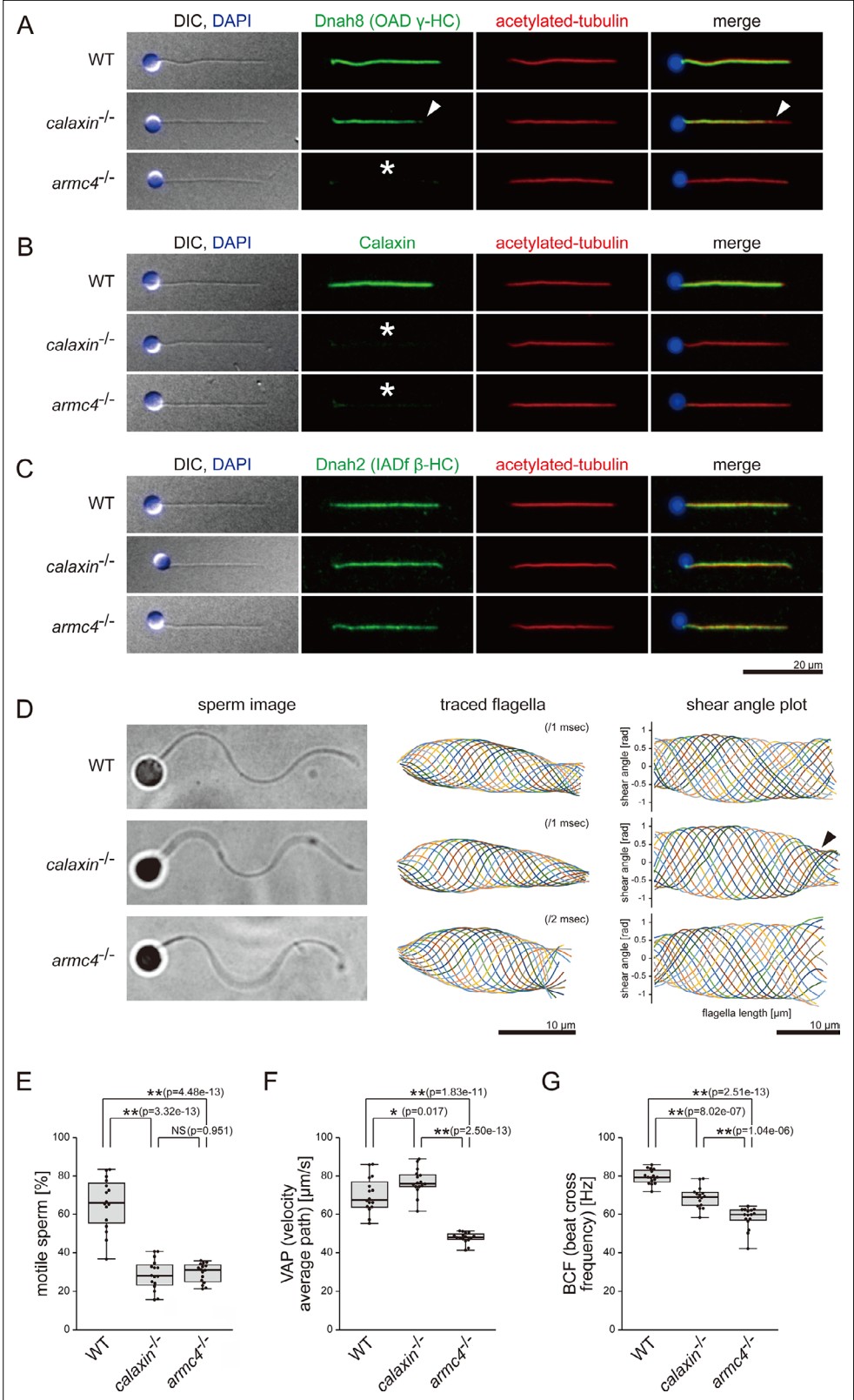

**Figure 2.** Mutations of *calaxin* and *armc4* cause loss of OAD in sperm flagella. (**A–C**) Immunofluorescence microscopy of zebrafish spermatozoa. Scale bar: 20 µm. (**A**) Dnah8 was localized along the entire length of sperm flagella in WT. In *calaxin⁻/⁻*, Dnah8 was lost at the distal region of sperm flagella (white arrowheads). In *arm4⁻/⁻*, Dnah8 was lost (white asterisk). (**B**) Calaxin was localized along the entire length of sperm flagella in WT. In

*Figure 2 continued on next page*

*Figure 2 continued*

*calaxin*[-/-] and *arm4*[-/-], Calaxin was lost (white asterisks). (**C**) Dnah2 was localized along the entire length of sperm flagella in WT, *calaxin*[-/-], and *arm4*[-/-]. (**D**) Phase-contrast microscopy images of swimming spermatozoa (left column), traces of beating flagella (middle column), and shear angle plots of traced flagella (right column). Swimming spermatozoa were filmed using a high-speed camera at 1000 fps (frames per second). Shear angles were plotted against the distance from the flagellar base. In *calaxin*[-/-], shear angle plots lost their slopes at the distal region of the flagella (black arrowhead). Scale bars: 10 μm. (**E–G**) Motilities of swimming spermatozoa were filmed using a high-speed camera at 200 fps and analyzed by CASA modified for zebrafish. For each zebrafish line, more than 600 spermatozoa were analyzed in total with 16 technical replicates. Spermatozoa with less than 20 μm/s velocities were considered immotile. (**H**) Ratio of motile spermatozoa. (**I**) Velocity of spermatozoa on averaged paths. The averaged paths were drawn by connecting the points of averaged sperm positions of contiguous 33 frames. (**J**) Frequencies at which sperm heads crossed their averaged paths. Boxes correspond to the first and third quartiles, lines inside the boxes indicate the medians, and whiskers extend to the full range of the data. p-Values were calculated with Tukey-Kramer test.

The online version of this article includes the following source data and figure supplement(s) for figure 2:

**Source data 1.** Numerical data of *Figure 2E*.

**Source data 2.** Numerical data of *Figure 2F*.

**Source data 3.** Numerical data of *Figure 2G*.

**Figure supplement 1.** Immunofluorescence microscopy of OAD-HCs in zebrafish spermatozoa.

**Figure supplement 2.** Sperm trajectories by CASA.

resolution of the OAD structure to 18.1 Å (*Figure 3—figure supplement 2D*). Concerned about the slight flexibility of OAD-DMT connections, we performed the local refinement focusing on OAD HCs (*Figure 4A–B*) or DC-DMT (*Figure 4D*).

First, we compared the zebrafish OAD structure with the well-studied *Chlamydomonas* OAD model (PDB-7kzm; *Walton et al., 2021*). To compare the main OAD structures, we omitted the α-HC and DC linkers from the *Chlamydomonas* OAD model. Despite the evolutionary distance, the *Chlamydomonas* model fitted well into the map of zebrafish OAD, showing the conservation of OAD structures (*Figure 4A′–B′*; *Video 8*). One exception is the LC4 protein in *Chlamydomonas* (*Figure 4C*, red dotted circle). LC4 is a calcium sensor protein of *Chlamydomonas* OAD and binds to the tail domain of γ-HC (*King and Patel-King, 1995*; *Sakato et al., 2007*). The LC4 orthologue is not found in the vertebrate genome, which is consistent with the lack of corresponding structure in the zebrafish OAD map (*Figure 4C*, lower).

Second, we compared the zebrafish DC structure with other vertebrates. The DC structure was highly conserved among vertebrates, as the bovine DC model (PDB-7rro; *Gui et al., 2021*) fitted well into the map of zebrafish DC-DMT (*Figure 4D′*; *Video 9*). The bovine DC model shows that vertebrate DC is composed of four parts: (a) Calaxin, (b) the Armc4-TTC25 complex, (c) the proximal CCDC151/114, and (d) the distal CCDC151/114 (*Figure 4E*). Among the four parts, three (a, b, and d) work as linkers between OAD and DMT, while (c) the proximal CCDC151/114 is embedded in the cleft between protofilaments of the DMT. Note that CCDC151/114 coiled-coil stretches across the two OAD repeats. The coiled-coil overlaps as its N-terminus region passes between Calaxin and the neighboring distal CCDC151/114 (*Figure 4D′*, red dotted circle).

To visualize the missing component in *calaxin*[-/-] OAD, we generated a difference map by subtracting *calaxin*[-/-] structure from WT structure (*Figure 4F′*, red; *Video 10*). *calaxin*[-/-] OADs were attached to DMT through only two linker structures (*Figure 4F*). The

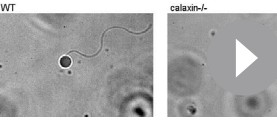

**Video 2.** Waveforms of swimming spermatozoa in WT, *calaxin*[-/-], and *armc4*[-/-]. Typical swimming spermatozoa, filmed by a high-speed camera at 1000 fps and played at 30 fps. Scale bar: 10 μm.

https://elifesciences.org/articles/84860/figures#video2

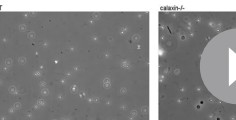
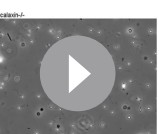
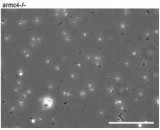

**Video 3.** Motilities of swimming spermatozoa in WT, *calaxin*[-/-], and *armc4*[-/-]. Typical movies of swimming spermatozoa for CASA, filmed by a high-speed camera at 200 fps and played at 30 fps. Scale bar: 100 μm.

https://elifesciences.org/articles/84860/figures#video3

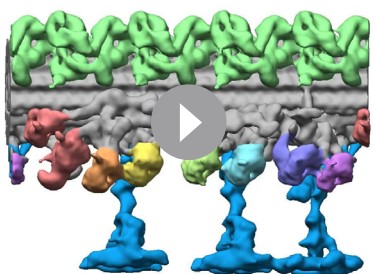

**Video 4.** Cryo-ET structure of WT DMT.
https://elifesciences.org/articles/84860/figures#video4

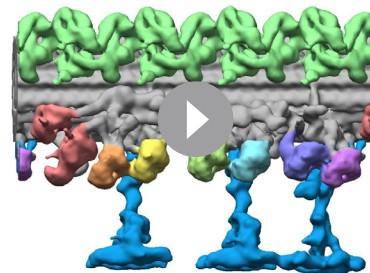

**Video 5.** Cryo-ET structure of *calaxin*⁻/⁻ DMT (OAD+ class).
https://elifesciences.org/articles/84860/figures#video5

major difference was observed only in the Calaxin region (*Figure 4F'*, white arrowhead), indicating that OADs kept mostly normal conformations even without Calaxin. However, the difference map also showed an additional missing structure adjacent to Calaxin (*Figure 4F'*, black arrowhead). When fitting the bovine DC model, this structure overlapped the N-terminus region of CCDC151/114, indicating that Calaxin can affect the conformation of neighboring DC components.

To test the interaction between Calaxin and other DC components, we performed an in vitro rescue experiment with recombinant Calaxin protein. We incubated the *calaxin*⁻/⁻ axonemes with purified Calaxin protein and then observed the OAD structure by cryo-ET. Interestingly, the Calaxin protein autonomously bound to the axoneme and rescued the deficient DC structure (*Figure 4G*). A difference map visualized the binding of recombinant Calaxin protein (*Figure 4G'*, white arrowhead) and the re-stabilized structure of the neighboring region of CCDC151/114 (*Figure 4G'*, black arrowhead). Ectopic binding of the Calaxin protein was not observed.

To assess the specificity of Calaxin binding, we also performed a rescue experiment with mEGFP-Calaxin (*Figure 4H–I*; *Figure 4—figure supplement 2*). *Ciona* Calaxin was reported to interact with β-tubulin (*Mizuno et al., 2009*), suggesting the possible binding of Calaxin along the entire length of the axoneme. However, the rescued axonemes showed partial loss of EGFP signal (*Figure 4H*, white arrowheads). This pattern resembled the OAD localization of *calaxin*⁻/⁻ in immunofluorescence microscopy, suggesting the preferential binding of Calaxin to the remaining OAD-DC. mEGFP alone showed no interaction with the axoneme (*Figure 4H*, asterisk). A difference map confirmed the binding of mEGFP-Calaxin to DC, as an additional density of mEGFP was observed adjacent to the Calaxin structure (*Figure 4I*).

## Calaxin stabilizes the docking of OAD onto DMT

*calaxin*⁻/⁻ spermatozoa showed loss of distal OAD with unclear boundaries (*Figure 2A*, white arrowhead). To assess the detailed distribution of *calaxin*⁻/⁻ OADs, we classified the 24 nm repeat unit of DMT into OAD+ and OAD- classes (*Figure 5A*). A slice

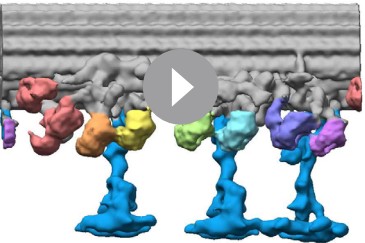

**Video 6.** Cryo-ET structure of *calaxin*⁻/⁻ DMT (OAD- class).
https://elifesciences.org/articles/84860/figures#video6

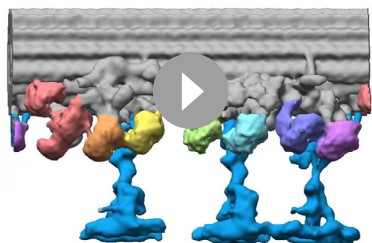

**Video 7.** Cryo-ET structure of *armc4*⁻/⁻ DMT.
https://elifesciences.org/articles/84860/figures#video7

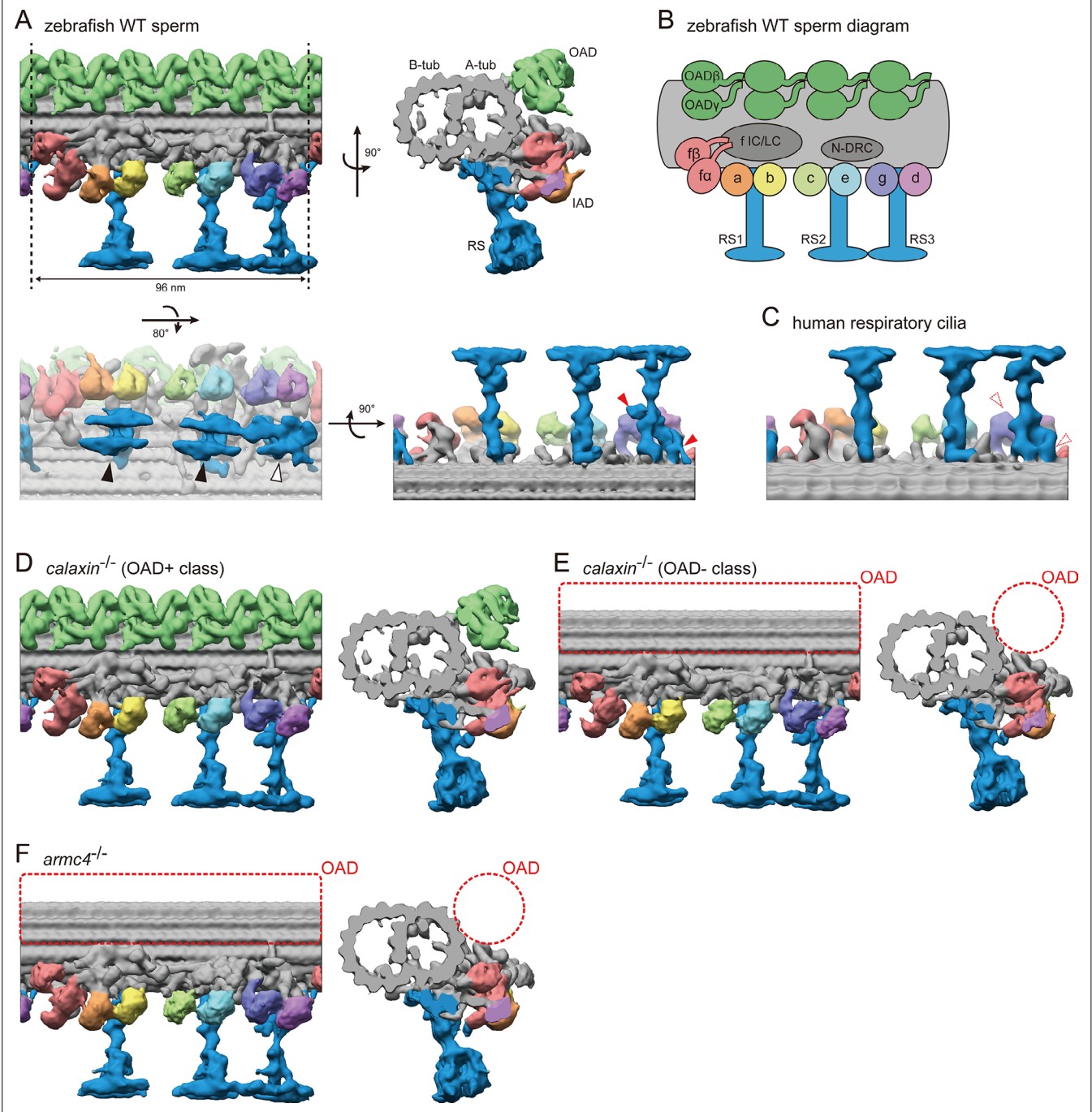

**Figure 3.** Cryo-ET structures of DMTs in WT and mutant sperm flagella. (**A**) DMT structure of WT zebrafish sperm flagella. A-tub and B-tubule of DMT, respectively. OAD: outer arm dynein, IAD: inner arm dynein, RS: radial spoke. Upper left: side view, upper right: base-to-tip view, lower left: bottom view, and lower right: back view of RSs. Black arrowheads indicate the spoke heads of RS1 and RS2. White arrowhead indicates the spoke head of RS3. Red arrowheads indicate the RS3 structures which are not found in the RS3 in human respiratory cilia. (**B**) Diagram of DMT structure. N-DRC: nexin-dynein regulatory complex. f IC/LC: IAD-f intermediate chain and light chain complex. (**C**) Back view of RSs in human respiratory cilia (EMD-5950; *Lin et al., 2014*). (**D–E**) DMT structures of *calaxin⁻/⁻* sperm flagella. Structural classification sorted the subtomograms into two classes: (**D**) OAD+ class (70.7%, 6122 particles) and (**E**) OAD- class (29.3%, 2535 particles). (**F**) DMT structure of *armc4⁻/⁻* sperm flagella. Green: OAD, pale red: IAD-f, orange: IAD-a, yellow: IAD-b, light-green: IAD-c, cyan: IAD-e, indigo: IAD-g, violet: IAD-d, and blue: RSs. Red circles indicate the loss of OAD.

The online version of this article includes the following source data and figure supplement(s) for figure 3:

**Figure supplement 1.** Computational process to generate combined volumes.

**Figure supplement 2.** Fourier shell correlations of averaged subtomograms.

**Figure 3-figure supplement 2-source data 1** .Particle numbers and resolutions of each averaged subtomogram.

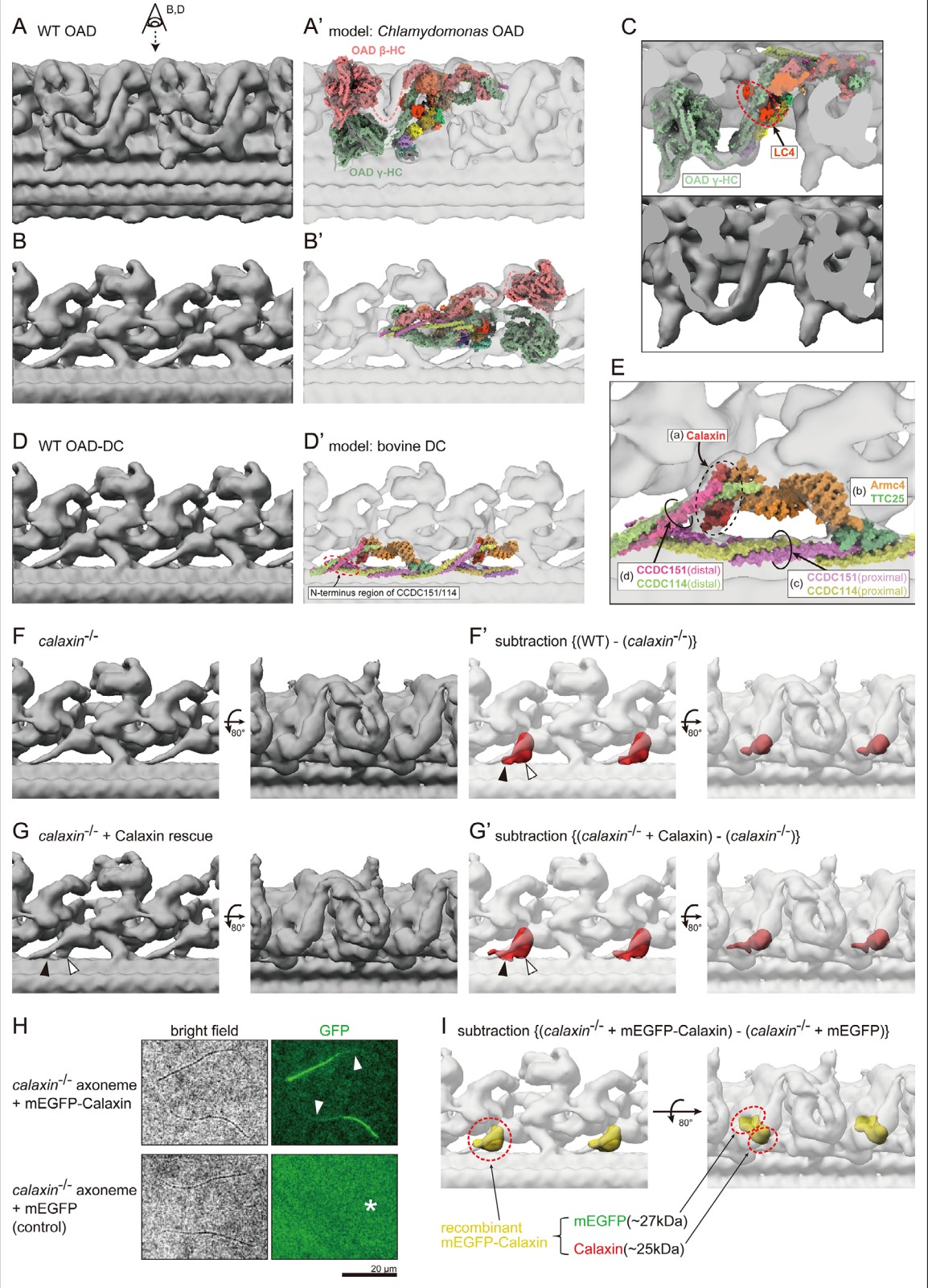

**Figure 4.** Cryo-ET structures of OADs in WT and *calaxin*⁻/⁻ sperm flagella. (**A and B**) OAD structure of WT zebrafish sperm flagella. Local refinement was performed focusing on OAD HCs. (**B**) shows the top view of A (eye and arrow). (**A' and B'**) Comparison of zebrafish OAD structure with *Chlamydomonas* OAD model (PDB-7kzm; *Walton et al., 2021*). α-HC and DC linkers were omitted from the *Chlamydomonas* OAD model. (**C**) Detailed structure around OAD γ-HC. Upper: composite image of zebrafish OAD and *Chlamydomonas* OAD model. Lower: zebrafish OAD only. *Chlamydomonas* OAD has

*Figure 4 continued on next page*

*Figure 4 continued*

the LC4 protein (red circle) attached to the γ-HC tail. (**D**) OAD-DC structure of WT zebrafish sperm flagella. Local refinement was performed focusing on DC. (**D′**) Comparison of zebrafish DC structure with bovine DC model (PDB-7rro; *Gui et al., 2021*). Red dotted circle indicates the N-terminus region of the CCDC151/114. (**E**) Detailed structure of DC. DC is composed of four linker structures: (**a**) Calaxin, (**b**) the Armc4-TTC25 complex, (**c**) the proximal CCDC151/114, and (**d**) the distal CCDC151/114. (**F**) OAD-DC structure of *calaxin⁻/⁻* sperm flagella. (**F′**) Composite image of difference map (red; subtraction of F from WT) and *calaxin⁻/⁻* OAD-DC (translucent). Difference map shows the densities corresponding to the Calaxin (white arrowhead) and the adjacent CCDC151/114 (black arrowhead). (**G**) OAD-DC structure of *calaxin⁻/⁻* incubated with recombinant Calaxin protein. (**G′**) Composite image of difference map (red; subtraction of *calaxin⁻/⁻* from G) and *calaxin⁻/⁻* OAD-DC (translucent). (**H**) *calaxin⁻/⁻* sperm axoneme was incubated with recombinant proteins of mEGFP-Calaxin or mEGFP (control). Upper row: mEGFP-Calaxin binds to the limited region of *calaxin⁻/⁻* axoneme, with the partial loss of EGFP signals (white arrowheads). Lower row: mEGFP has no interaction with *calaxin⁻/⁻* axoneme (asterisk). Scale bar: 20 µm. (**I**) Composite image of difference map (yellow; subtraction of *calaxin⁻/⁻* incubated with mEGFP from *calaxin⁻/⁻* rescued by mEGFP-Calaxin) and *calaxin⁻/⁻* OAD-DC (translucent). Difference map shows the densities of mEGFP and Calaxin.

The online version of this article includes the following source data and figure supplement(s) for figure 4:

**Source data 1.** Original SDS-PAGE image of recombinant proteins.

**Source data 2.** Annotated SDS-PAGE images of recombinant proteins.

**Figure supplement 1.** Comparison of OAD-DC structures in vertebrates and *Chlamydomonas*.

**Figure supplement 2.** Interactions of recombinant Calaxin protein with WT and mutant sperm axonemes.

of raw tomogram confirmed that particles in OAD+ class contain the OAD density (*Figure 5B*, blue circles), which is not observed in the OAD- class particles (*Figure 5B*, red circles). We schematically displayed the distributions of two OAD classes using colored grids as individual particles in the tomogram (*Figure 5C–D*). Note that each tomogram contains only ~3 µm of the axoneme, while the total length of sperm flagella is ~30 µm. In WT, almost all particles were sorted into OAD+ class (*Figure 5C*). In *calaxin⁻/⁻*, the ratio of OAD+ class to OAD- class varied among tomograms (*Figure 5D*), reflecting the different distances from the sperm head. Analysis of detailed OAD distributions along *calaxin⁻/⁻* axoneme revealed that OAD loss occurred even in the proximal part of the flagella (*Figure 5—figure supplement 1D*). Moreover, nine DMTs showed no obvious heterogeneity about the distribution of two OAD classes, suggesting that OAD and Calaxin are localized uniformly on all DMTs.

To test whether Calaxin stabilizes the docking of OADs, we examined the amount of OADs retained on the *calaxin⁻/⁻* axonemes (*Figure 5E*; *Figure 5—figure supplement 2*). We incubated purified *calaxin⁻/⁻* axonemes with or without Calaxin proteins, in buffers containing 50 mM K-acetate as a basis and additional NaCl with different concentrations. Higher NaCl concentrations (200, 250, or 300 mM) caused drastic loss of retained OADs in both Calaxin-conditions. However, at lower NaCl concentrations (0, 50, or 100 mM), axonemes incubated with Calaxin retained more OADs than the axonemes without Calaxin, showing that Calaxin contributes to stabilizing the OAD-DMT interaction (*Figure 5E*, bottom). Importantly, even in buffers with 50 mM K-acetate and 0 mM NaCl, which are generally used to preserve the intact OAD-DMT, *calaxin⁻/⁻* axonemes reduced the amount of OADs. This data suggests that the OAD docking in *calaxin⁻/⁻* was so unstable that further OAD dissociations occurred during the incubation.

Taken together, we present a model of the stabilizing process of OAD docking onto DMT (*Figure 5F*). OADs can bind to DMT through two linker structures, which are composed of CCDC151, CCDC114, TTC25, and Armc4 (*Figure 4E*). The two linkers are insufficient to stabilize the OAD

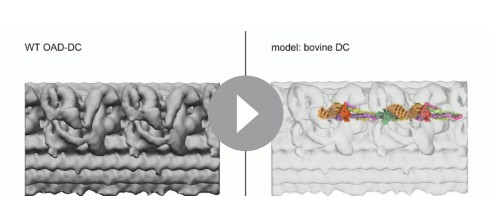

**Video 8.** Cryo-ET structure of WT OAD. Left: OAD structure of WT zebrafish sperm flagella. Right: Comparison of zebrafish OAD structure with *Chlamydomonas* OAD model (PDB-7kzm; *Walton et al., 2021*). α-HC and DC linkers were omitted from the *Chlamydomonas* OAD model.

https://elifesciences.org/articles/84860/figures#video8

**Video 9.** Cryo-ET structure of WT OAD-DC. Left: OAD-DC structure of WT zebrafish sperm flagella. Right: Comparison of zebrafish DC structure with bovine DC model (PDB-7rro; *Gui et al., 2021*).

https://elifesciences.org/articles/84860/figures#video9

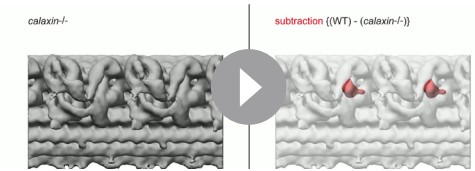

**Video 10.** Cryo-ET structure of *calaxin*[-/-] OAD-DC. Left: OAD-DC structure of *calaxin*[-/-] sperm flagella. Right: Composite of difference map (red; subtraction of *calaxin*[-/-] from WT) and *calaxin*[-/-] OAD-DC (translucent).
https://elifesciences.org/articles/84860/figures#video10

docking and cause the loss of OADs at various regions of the axoneme. Calaxin protein can bind to the pre-existing DC and stabilizes the OAD-DMT interactions as the third linker structure. Although Calaxin was initially identified as a calcium sensor of OAD in *Ciona* spermatozoa, our study illustrates the additional function of Calaxin as an important factor for stable OAD-DMT docking.

## Calaxin requires Armc4 to be localized to cilia

Although the above structural analyses showed that Calaxin stabilizes the OAD docking, it is unclear whether Calaxin is transported together with other DC components. To test the idea, we performed immunofluorescence microscopy of multiciliated cells of zebrafish olfactory epithelium (*Figure 6*). Axonemal components larger than ~50 kDa undergo gated entry into the ciliary compartment after synthesis in the cytoplasm (*Kee et al., 2012*; *Takao and Verhey, 2016*). Multiciliated cells are suitable for distinguishing proteins localized to the cytoplasm, ciliary base, and ciliary compartments.

Acetylated-tubulin signals show the ciliary compartment (*Figure 6*, red), while DAPI signals indicate the nuclei and the surrounding cytoplasmic regions (*Figure 6*, blue). In WT, both Dnah8 and Calaxin were localized in the ciliary compartment (*Figure 6A*). Dnah8 signals were also observed at the boundary region between the ciliary compartment and cytoplasm, suggesting that OAD components were accumulated at the ciliary base before transport into cilia. In *calaxin*[-/-], Calaxin was absent (*Figure 6B*, asterisks), but Dnah8 was localized in the ciliary compartment (*Figure 6B*). In *armc4*[-/-], both Dnah8 and Calaxin were absent from the ciliary compartment (*Figure 6C*). Twister (also known as Pih1d3/Dnaaf6) is one of the DNAAFs and is required for the cytoplasmic OAD pre-assembly. Previously, we analyzed the phenotype of *twister*[-/-] zebrafish and revealed the complete loss of OAD from the mutant axoneme (*Yamaguchi et al., 2018*). Consistent with our previous report, *twister*[-/-] showed the absence of Dnah8 from the ciliary compartment (*Figure 6D*). In contrast, Calaxin remained in the ciliary compartment of *twister*[-/-] cells, indicating that the ciliary localization of Calaxin is independent of OAD. Intriguingly, *armc4*[-/-] cells showed ectopic Calaxin accumulation at the ciliary base (*Figure 6C*, white arrowheads). The small molecular weight of Calaxin (~25 kDa) suggests the possible diffusional entry of Calaxin into the ciliary compartment. However, cytoplasmic accumulation of Calaxin in *armc4*[-/-] cells indicates that Calaxin requires Armc4 to be localized to cilia, implying the transport of DC components as pre-assembled complexes.

## Vertebrate DC exhibits slight conformation change in the $Ca^{2+}$ condition

Calaxin modulates the OAD activity in response to the $Ca^{2+}$ increase (*Mizuno et al., 2012*). To test whether $Ca^{2+}$ induces the change of OAD-DC conformation, we compared the structures of OAD-DC in different $Ca^{2+}$ conditions: 1 mM EGTA (for $Ca^{2+}$-free; *Figure 7A–A′*) and 1 mM $Ca^{2+}$ (*Figure 7B–B′*). The structures shown in *Figure 7A and A′* are the same as those in *Figure 4D and F′*, respectively, since the experiments so far were performed in 1 mM EGTA condition. Although no noticeable difference was observed in the OAD core region, DC exhibited slight conformation changes between $Ca^{2+}$ conditions. In 1 mM EGTA condition, the density overlapping the N-terminus region of CCDC151/114 was observed (*Figure 7A–A′*, black arrowheads), which was not found in 1 mM $Ca^{2+}$ condition. In 1 mM $Ca^{2+}$ condition, an additional density appeared around DC (*Figure 7B–B′*, white arrowheads), which was not found in 1 mM EGTA condition. The corresponding structure to the additional density was not identified in the bovine DC model. We also performed a rescue experiment of *calaxin*[-/-] OADs in 1 mM $Ca^{2+}$ condition, since *Ciona* Calaxin was reported to switch its interactor depending on $Ca^{2+}$ concentration (*Mizuno et al., 2009*; *Inaba, 2015*). However, we found the binding of Calaxin to DC (*Figure 7C–C′*), indicating that the Calaxin-DC association is independent of $Ca^{2+}$ conditions.

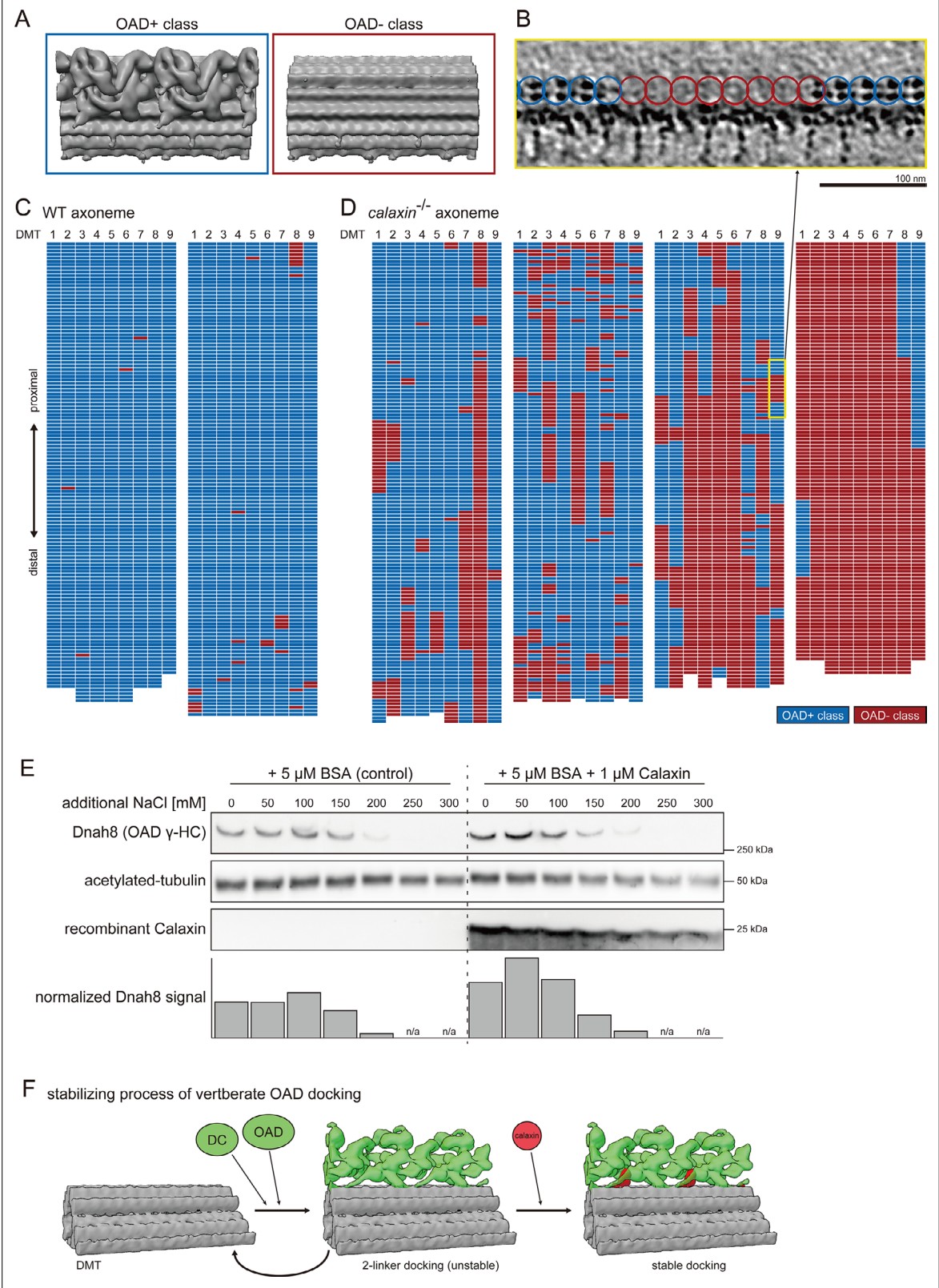

**Figure 5.** Calaxin is required for the stable docking of OAD onto DMT. (**A–D**) Structural classification sorted the 24 nm repeat units of DMT into OAD+ class (blue) and OAD- class (red). (**A**) Averaged structures of each class. (**B**) Tomographic slice shows the side view of DMT. Blue or red circles indicate the class of each 24 nm repeat unit of DMT. Scale bar: 100 nm. (**C–D**) Distribution patterns of each class. 24 nm repeat units on each DMT are schematically displayed as individual colored grids. (**C**) Two typical tomograms of WT axoneme. (**D**) Four typical tomograms of *calaxin*⁻/⁻ axoneme.

*Figure 5 continued on next page*

*Figure 5 continued*

Yellow square indicates the region displayed in B. (**E**) Immunoblot of *calaxin*[-/-] sperm axonemes incubated with or without recombinant Calaxin protein in different salt concentration buffers. Bottom row shows the bar graph of Dnah8 signals, normalized by the amount of acetylated tubulin signals. (**F**) Model of the stabilizing process of vertebrate OAD docking onto DMT.

The online version of this article includes the following source data and figure supplement(s) for figure 5:

**Source data 1.** Original blot images of *Figure 5E*, Dnah8.

**Source data 2.** Original blot images of *Figure 5E*, acetylated tubulin and recombinant Calaxin.

**Source data 3.** Annotated blot images of *Figure 5E*, Dnah8.

**Source data 4.** Annotated blot images of *Figure 5E*, acetylated tubulin and recombinant Calaxin.

**Figure supplement 1.** Detailed distribution of OADs in *calaxin-/-* sperm axoneme.

**Figure supplement 2.** Experimental replication confirmed the reproducibility of data shown in *Figure 5E*.

**Figure supplement 2—source data 1.** Original blot images of *Figure 5—figure supplement 2*, Dnah8.

**Figure supplement 2—source data 2.** Original blot images of *Figure 5—figure supplement 2*, acetylated tubulin.

**Figure supplement 2—source data 3.** Original blot images of *Figure 5—figure supplement 2*, recombinant Calaxin.

**Figure supplement 2—source data 4.** Annotated blot images of *Figure 5—figure supplement 2*.

## Discussion

In this study, we analyzed the functions of two DC components in vertebrates: Calaxin and Armc4, and demonstrated that each component contributes differently to the OAD-DMT interaction. Calaxin was first identified as a $Ca^{2+}$-dependent OAD regulator in *Ciona* spermatozoa. Our data revealed a novel function of Calaxin as an important factor for stable OAD-DMT docking. OADs in *calaxin*[-/-] were tethered to DMT through only two linker structures, which were insufficient to stabilize the OAD docking. Calaxin protein rescued the instability of OAD docking as the third linker structure, suggesting the stabilizing process of OAD-DMT docking in vertebrates.

### *calaxin* mutation causes partial loss of OAD, unlike other DC component genes

*calaxin*[-/-] spermatozoa exhibited a unique OAD distribution, with OAD-missing clusters at various regions of the flagella. Interestingly, OADs decreased gradually toward the distal end, by which the mechanism is unclear. The axoneme is elongated by adding flagellar components to its distal end during ciliogenesis (*Johnson and Rosenbaum, 1992*). IFT88, a component of the IFT machinery, disappears as the spermatozoa mature (*San Agustin et al., 2015*). Thus, we speculate that the OAD supply at the distal sperm axoneme is insufficient to compensate for the OAD dissociation in the *calaxin*[-/-]. Consistent with this idea, distal OAD loss is the sperm-specific phenotype, as olfactory epithelial cells in *calaxin*[-/-] have Dnah8 along the entire length of the cilia (*Figure 6B*).

In mouse *Calaxin*[-/-] mutant, motile cilia in various organs (sperm flagella, tracheal cilia, and brain cilia) showed abnormal motilities, although OADs in the mutant cilia/flagella seemed mostly intact when observed by conventional transmission electron microscopy (*Sasaki et al., 2019*). In our study, however, we revealed that mutation of zebrafish *calaxin* caused OAD-missing clusters at various regions of the flagella, by using detailed cryo-ET analysis and immunofluorescence microscopy. Thus, we speculate that the same OAD defects to zebrafish *calaxin*[-/-] caused abnormal ciliary motilities in mouse *Calaxin*[-/-] mutant. One exception is the mouse nodal cilia. In mouse *Calaxin*[-/-] mutant, the formation of nodal cilia was significantly disrupted (*Sasaki et al., 2019*). On the other hand, zebrafish *calaxin*[-/-] mutant showed the normal formation of Kupffer's vesicle cilia (orthologous to the mouse nodal cilia), suggesting the tissue-specific function of Calaxin on the ciliary formation.

Unlike *calaxin*, mutations of other DC component genes cause drastic loss of OAD. Human PCD patients with a mutation of *CCDC151*, *CCDC114*, or *TTC25* lose DNAH5 (human OAD γ-HC) from the entire length of respiratory cilia (*Hjeij et al., 2014*; *Onoufriadis et al., 2013*; *Wallmeier et al., 2016*). Zebrafish *armc4* mutants showed complete loss of OAD in the sperm flagella and olfactory epithelial cilia. Intriguingly, however, human patients with *ARMC4* mutation lose DNAH5 from only the distal region of the respiratory cilia (*Hjeij et al., 2013*).

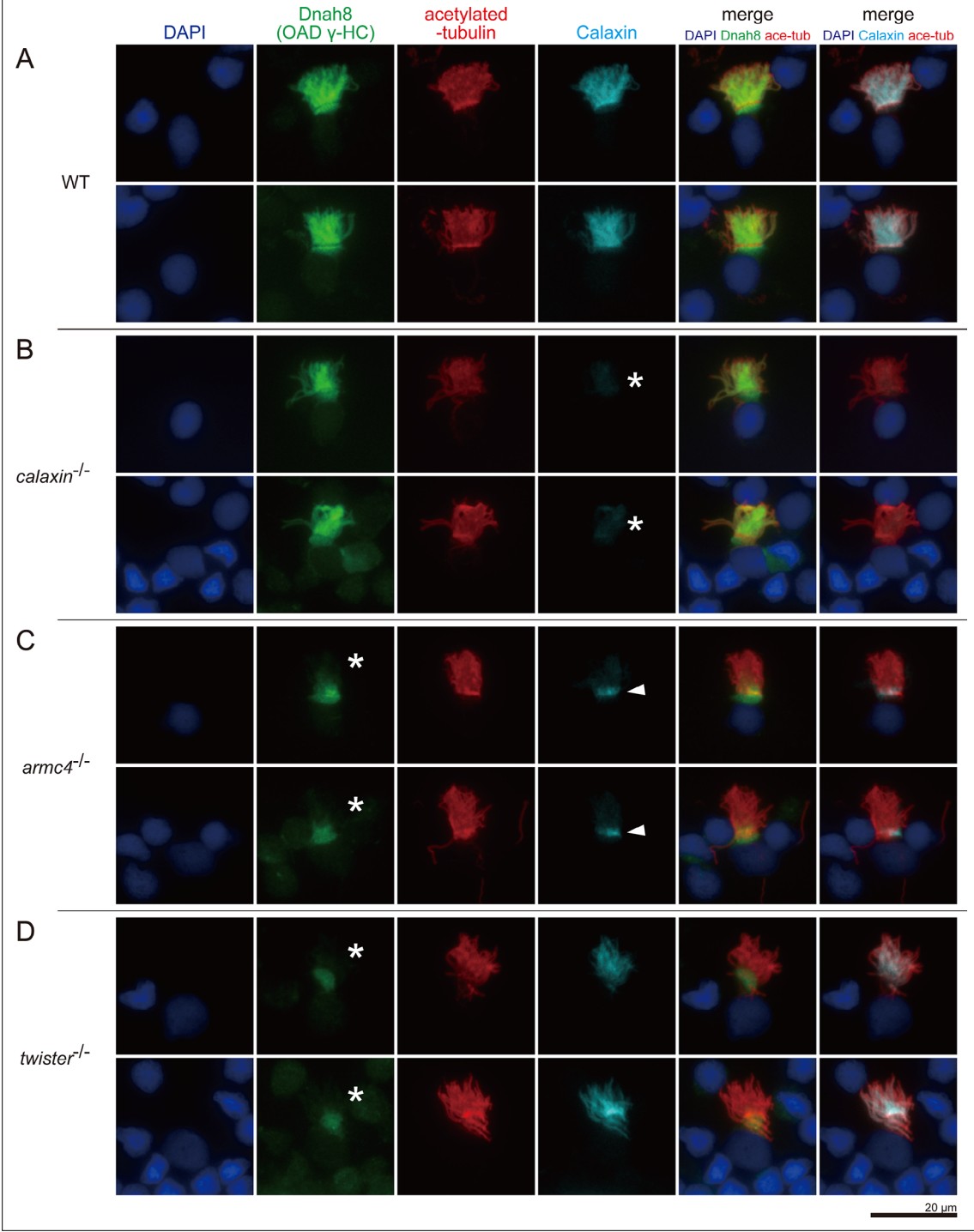

**Figure 6.** Calaxin requires Armc4 to be localized to cilia. (**A–D**) Immunofluorescence microscopy of multiciliated cells of zebrafish olfactory epithelium. (**A**) WT. (**B**) *calaxin*⁻/⁻. Calaxin signal was lost (white asterisks). (**C**) *armc4*⁻/⁻. Ciliary localization of Dnah8 was lost (white asterisks). Calaxin was accumulated at the ciliary base (white arrowheads). (**D**) *twister*⁻/⁻. Ciliary localization of Dnah8 was lost (white asterisks). Scale bar: 20 μm.

One possible explanation for the discrepancy of *armc4*⁻/⁻ phenotypes is the difference in OAD composition between humans and zebrafish. Human respiratory cilia have two OAD types: the proximal DNAH11/DNAH5-containing OAD (type-1) and the distal DNAH9/DNAH5-containing OAD (type-2) (*Fliegauf et al., 2005*; *Dougherty et al., 2016*). DNAH11 and DNAH9 are the OAD β-HC isoforms. Zebrafish spermatozoa have Dnah9/Dnah8 along the entire length of flagella (*Figure 2—figure*

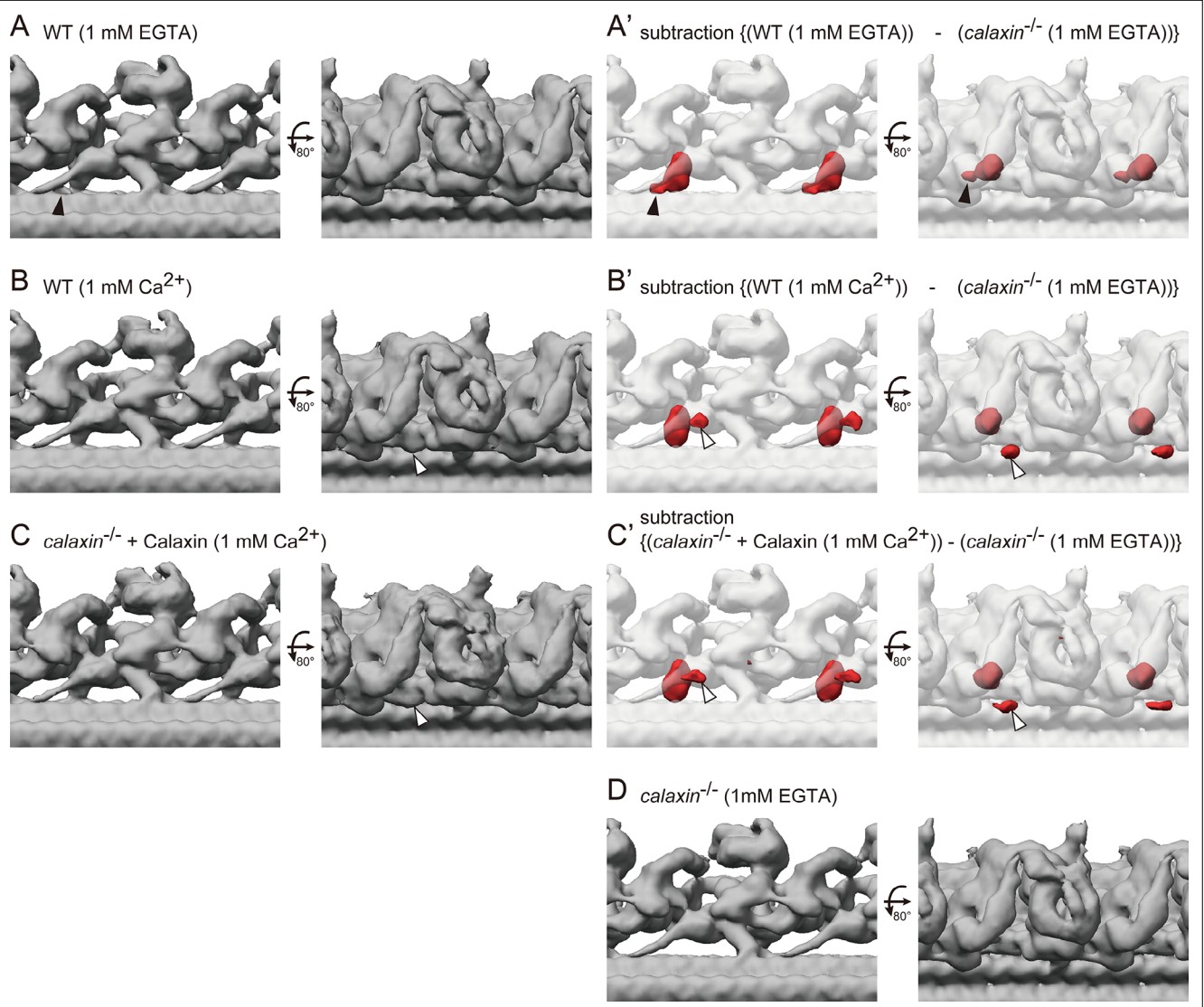

**Figure 7.** Cryo-ET structures of OADs in different $Ca^{2+}$ conditions. (**A–B**) OAD-DC structures of WT sperm flagella in different $Ca^{2+}$ conditions: (**A**) 1 mM EGTA condition (for $Ca^{2+}$-free) and (**B**) 1 mM $Ca^{2+}$ condition. (**C**) OAD-DC structure of $calaxin^{-/-}$ sperm flagella incubated with recombinant Calaxin protein in 1 mM $Ca^{2+}$ condition. (**A'-C'**) Composite images of difference map (red; subtraction of $calaxin^{-/-}$ in 1 mM EGTA condition from A, B, and C, respectively) and $calaxin^{-/-}$ OAD-DC (translucent). Black arrowheads in A indicate the CCDC151/114 structure adjacent to Calaxin, which is not observed in B and C. White arrowheads in B and C indicate the additional density around DC, which is not observed in A. (**D**) OAD-DC structure of $calaxin^{-/-}$ sperm in 1 mM EGTA condition, which was used to generate (**A'-C'**).

The online version of this article includes the following figure supplement(s) for figure 7:

**Figure supplement 1.** Student's t-test to compare the WT and $calaxin^{-/-}$ OAD-DC structures.

*supplement 1*). Thus, distal specific loss of DNAH5 in human *ARMC4* patients can be correlated with the loss of type-2 OADs.

Moreover, mutation positions differ between human *ARMC4* patients and our zebrafish mutant. Most human *ARMC4* patients have mutations in the C-terminus regions, suggesting the existence of truncated ARMC4 protein. Actually, immunofluorescence microscopy detected the ARMC4 signals in the patient's respiratory cells (*Hjeij et al., 2013*). It is possible that the truncated ARMC4 partially incorporates the type-1 OADs into the proximal respiratory cilia. In contrast, our zebrafish *armc4*$^{-/-}$ has a stop-codon in the exon2, which leaves only 41 amino acids for possible encoding, while zebrafish Armc4 has 1047 amino acids.

## DC components have different dependencies for ciliary localization

The ciliary localization of Calaxin was Armc4-dependent, suggesting the ciliary targeting of DC components as pre-assembled complexes (*Figure 6C*). In contrast, the ciliary localization of Armc4, CCDC151, CCDC114, and TTC25 was independent of Calaxin, because the DC components except Calaxin were observed in the *calaxin*$^{-/-}$ axoneme by cryo-ET. Previous studies also reported the different dependencies of DC components for ciliary localization: ARMC4 depends on CCDC114 for its ciliary localization, while CCDC114 localization is independent of ARMC4 (*Hjeij et al., 2013*). TTC25 is required for the ciliary localization of CCDC151, CCDC114, and ARMC4, but TTC25 is localized in cilia independent of these DC components (*Wallmeier et al., 2016*). Importantly, TTC25 interacts with IFT complexes (*Xu et al., 2015*). Thus, we hypothesize that DC components are loaded to IFT in the order of TTC25, CCDC151/114, Armc4, and Calaxin, to be transported into the ciliary compartment. The cytoplasmic behavior of DC components remains elusive, compared to the emerging mechanisms of the cytoplasmic OAD pre-assembly (*King, 2021*; *Lee et al., 2020*). Further analysis of the cytoplasmic localization and interaction of DC components could contribute to understanding the mechanism of the ciliary targeting of DC complexes.

## Ca$^{2+}$ induces the conformation change of vertebrate DC

In *Chlamydomonas*, Ca$^{2+}$ induces the conformation change of OAD γ-HC, which is correlated with the LC4 function (*Sakato et al., 2007*). LC4 is a calcium sensor protein of *Chlamydomonas* OAD and binds to the γ-HC tail domain. Vertebrates lack the LC4 orthologue but have Calaxin as a calcium sensor of OAD, suggesting the diverse mechanisms of the Ca$^{2+}$-dependent OAD regulation (*Inaba, 2015*). Our cryo-ET analysis confirmed the lack of LC4 structure in the vertebrate OAD (*Figure 4C*). Consistent with this, Ca$^{2+}$ induced no noticeable difference in the vertebrate OAD core region (*Figure 7A–B*).

On the other hand, Ca$^{2+}$ induced several conformation changes in the vertebrate DC structure, which can be correlated with the Calaxin function. First, Ca$^{2+}$ changed the density overlapping the N-terminus region of CCDC151/114 (*Figure 7A*, black arrowheads). The change in the same region was observed by the binding of Calaxin protein (*Figure 4F–G*, black arrowheads). Second, we found an additional density around the DC in the Ca$^{2+}$ condition (*Figure 7B–C*, white arrowheads). Although the corresponding structure to the additional density is not identified in the bovine DC model, it is close to the distal region of CCDC151/114, which interacts with Armc4. Calaxin can affect the conformation of these components, since the bovine DC model shows that Calaxin binds at the interface between Armc4 and the distal region of CCDC151/114 (*Figure 4E*). Thus, we speculate that Calaxin, as a calcium sensor, modulates the conformation of other DC components in the Ca$^{2+}$ condition.

Regarding the Calaxin conformation, a previous biochemical analysis reported that *Ciona* Calaxin switches its interactor depending on Ca$^{2+}$: β-tubulin at lower Ca$^{2+}$ concentration and OAD γ-HC at higher Ca$^{2+}$ concentration (*Mizuno et al., 2009*). Moreover, a crystal structure analysis revealed the conformational transition of *Ciona* Calaxin toward the closed state by Ca$^{2+}$-binding (*Shojima et al., 2018*). In this study, however, such conformation change of Calaxin was not detected, probably due to insufficient resolution of our cryo-ET analysis. More detailed structural analyses in the Ca$^{2+}$ condition are required to understand the mechanism of the Ca$^{2+}$-dependent OAD regulation.

## Materials and methods

### Key resources table

| Reagent type (species) or resource | Designation | Source or reference | Identifiers | Additional information |
|---|---|---|---|---|
| Gene (*Danio rerio*) | *calaxin* | NA | ZFIN: ZDB-GENE-040914–40 | also known as *efcab1*; *odad5* |
| Gene (*Danio rerio*) | *armc4* | NA | ZFIN: ZDB-GENE-100316–7 | also known as *odad2* |
| Antibody | Anti-acetylated tubulin (mouse monoclonal) | Sigma-Aldrich | Sigma-Aldrich: T6793 | 1:500 in immunofluorescence microscopy; 1:5000 in immunoblot analysis |

*Continued on next page*

*Continued*

| Reagent type (species) or resource | Designation | Source or reference | Identifiers | Additional information |
|---|---|---|---|---|
| Antibody | Anti-Dnah8 (rabbit polyclonal) | PMID:29741156 | | against aa 895–1402; 1:50 in immunofluorescence microscopy; 1:400 in immunoblot analysis |
| Antibody | Anti-Dnah2 (rabbit polyclonal) | PMID:29741156 | | against aa 802–1378; 1:50 in immunofluorescence microscopy; 1:400 in immunoblot analysis |
| Antibody | Anti-Calaxin (guinea pig polyclonal) | PMID:31240264 | | against full-length; 1:50 in immunofluorescence microscopy; 1:400 in immunoblot analysis |
| Antibody | Anti-Dnah9 (rabbit polyclonal) | This paper | | against aa 535–1002; 1:50 in immunofluorescence microscopy; 1:400 in immunoblot analysis |
| Software, algorithm | CASA modified for zebrafish | PMID:17137620 | | |
| Software, algorithm | SerialEM | PMID:16182563 | | |
| Software, algorithm | MotionCor2 | PMID:28250466 | | |
| Software, algorithm | IMOD | PMID:8742726 | | |
| Software, algorithm | PEET | PMID:16917055 | | |
| Software, algorithm | UCSF Chimera | PMID:15264254 | | |
| Software, algorithm | EMAN2 | PMID:16859925 | | |

## Zebrafish maintenance and genome-editing

The zebrafish breeding system was maintained at 28.5 °C on a 13.5 hr light /10.5 hr dark cycle. Embryos

**Table 1.** Oligonucleotide sequences used in this study.

| Purpose | Name | Sequence |
|---|---|---|
| Mutant generation | calaxingRNA.oligoF | ATTTAGGTGACACTATAGCGTCGGTCATCCCGAA AGTGGTTTTAGAGCTAGAAATAGCAAG |
| | armc4gRNA.oligoF | ATTTAGGTGACACTATAGTACTTCAGTGAGAGCC ACCGTTTTAGAGCTAGAAATAGCAAG |
| | constant.oligoR | AAAAGCACCGACTCGGTGCCACTTTTTCAAGTTG ATAACGGACTAGCCTTATTTTAACTTGCTATTTCT AGCTCTAAAAC |
| | calaxin_check.F | GGAGAGCAGGCAGAGAGAAAG |
| | calaxin_check.R | CTGCACTGCAAATTGTGATTG |
| | armc4_check.F | CTAGAGAACAGCCTCCTGAATA |
| | armc4_check.R | GTGAAATCAGACACTTCTAGAGAT |
| | EcoRI_calaxin.F | GGGAATTCCCATGCTGAAAATGTCGGCGATG |
| | EcoRI_calaxin.R | GGGAATTCTTATTCTTTGCAGTGTTCGTGTTTCTG |
| Recombinant Calaxin protein | mEGFP.F | ATGGTGAGCAAGGGCGAG |
| | mEGFPdel229.R | GATCCCGGCGGCGGTCAC |
| | pGEX6p2-mEGFP.R | GCCCTTGCTCACCATGGGAATTCCTGGGGATCC |
| | mEGFPdel229-calaxin.F | ACCGCCGCCGGGATCATGCTGAAAATGTCGGCGA |
| Dnah9 antigen | BamHI_dnah9.F | CGGGATCCGAGCAGCCGCTGATAGCA |
| | SalI_dnah9.R | CGGTCGACTTTGCGGTCGTCCACGTA |

and larvae were incubated at the same temperature in 1/3 Ringer's solution (39 mM NaCl, 0.97 mM KCl, 1.8 mM CaCl$_2$, and 1.7 mM HEPES, pH 7.2). Developmental stages of embryos and larvae are described according to hpf (hours post fertilization) at 28.5 °C and the morphological criteria (*Kimmel et al., 1995*). The generation of *calaxin*$^{-/-}$ zebrafish was described in the previous report (*Sasaki et al., 2019*; referred to as *efcab1*$^{-/-}$ zebrafish). The generation of *twister*$^{-/-}$ zebrafish was described in the previous report (*Yamaguchi et al., 2018*). For *armc4*$^{-/-}$ zebrafish, CRISPR/Cas9 genome-editing was performed according to the previously reported method (*Gagnon et al., 2014*), with the target site: GTACTTCAGTGAGAGCCACCCGG. Genomic DNA was extracted from the embryos, and the target loci were amplified to check the mutations by sanger-sequencing. Oligonucleotide sequences used in this study are summarized in *Table 1*. After identifying the founder fish, a homozygous mutant line was generated and maintained.

## Kupffer's vesicle cilia analysis

Embryos developing Kupffer's vesicles were selected at 12 hpf and dechorionated before observations. The orientations of embryos were aligned in 0.8% of low-gelling temperature agarose (A9045, Sigma-Aldrich) dissolved in 1/3 Ringer's solution. The motility of Kupffer's vesicle cilia was observed using an inverted microscope (DMI6000B, Leica) under the bright-field condition and a high-speed camera (HAS-L1, Detect) at 1000 fps (frames per second). The direction of heart looping was observed at 36 hpf.

## Sperm treatment

Zebrafish spermatozoa were expelled from the dissected testis and collected in Hank's buffer (137 mM NaCl, 5.4 mM KCl, 0.25 mM Na$_2$HPO$_4$, 0.44 mM KH$_2$PO$_4$, 1.3 mM CaCl$_2$, 1.0 mM MgSO$_4$, and 4.2 mM NaHCO$_3$). For the purification of sperm axonemes, sperm heads and membranes were removed by adding 2% Nonidet P-40 to Hank's buffer. Demembranated axonemes were collected by centrifugation (5000 *g*, 3 min), then resuspended in HMDEKAc buffer (30 mM HEPES at pH 7.2, 5 mM MgSO$_4$, 1 mM dithiothreitol, 1 mM EGTA, and 50 mM CH$_3$COOK).

## Immunofluorescence microscopy

Samples were prepared on the ight-well glass slide (TF0808, Matsunami) coated with 0.01% polyethyleneimine solution. For sperm samples, spermatozoa in Hank's buffer were applied to the well, and the attached spermatozoa were briefly demembranated using 1% Nonidet P-40 for 2 min. For multiciliated cell samples, zebrafish olfactory rosettes were dissected in PBSE (phosphate-buffered saline containing 1 mM EGTA) and pushed against the well to attach the epithelial cells to the slide. Samples were fixed with 2% paraformaldehyde/Hank's buffer (sperm) or with 4% paraformaldehyde/ PBSE (multiciliated cell) for 10 min at room temperature, followed by treatment with cold acetone and methanol (–20 °C). After rehydration with PBST (phosphate-buffered saline containing 0.1% Tween-20), specimens were incubated with the blocking buffer (2% normal goat serum and 1% cold fish gelatin in PBST). Immunostaining was performed with monoclonal anti-acetylated tubulin antibody (1:500 dilution) and polyclonal antibodies (1:50 dilution) as primary antibodies. Fluorescence-conjugated secondary antibodies (1:250 dilution) were used with 2.5 µg/ml DAPI for nuclear staining. Specimens were mounted with Fluoro-KEEPER Antifade Reagent (Nacalai tesque). Sperm samples were observed using a fluorescence microscope (BX60, Olympus) and a CCD camera (ORCA-R2, Hamamatsu). Multiciliated cell samples were observed using a fluorescence microscope system (BZ-X700, Keyence).

## Sperm motility analysis

Spermatozoa were kept inactive in cold Hank's buffer until analyzed and used within 1 hr of sperm collection. Spermatozoa were activated by adding an abundant amount of 1/5×Hank's buffer. Spermatozoa were prepared on the glass slides with 30 µm spacers (200A10, Kyodo giken chemical) and covered with coverslips to provide a consistent fluid depth. For the observation of free-swimming spermatozoa, 2 mg/ml of BSA (A9418, Sigma-Aldrich) was added to the buffers, which prevented spermatozoa from attaching to the glass surface. For sperm waveform analysis, phase-contrast images

of swimming spermatozoa were filmed using a microscope (BX53, Olympus) and a high-speed camera (Eosens MC1362, Mikrotron) at 1000 fps. Sperm heads were tracked with the MTrack2 plugin in ImageJ/Fiji (NIH) and overlaid. Traces of flagella waveforms were generated with the AnalyzeSkeleton plugin in ImageJ/Fiji. Shear angles were calculated as tangent angles of the traced flagella and plotted against the distance from the flagellar base. For CASA, phase-contrast images of swimming spermatozoa were filmed using an inverted microscope (DMI6000B, Leica) and a high-speed camera (HAS-L1, Detect) at 200 fps. CASA modified for zebrafish was performed as previously reported (*Wilson-Leedy and Ingermann, 2007*). Eight independent experiments with two times of 1 s observations were performed to obtain 16 technical replicates of CASA.

## Calaxin protein purification and rescue assays of *calaxin*[-/-] sperm axoneme

A full-length zebrafish *calaxin* sequence was subcloned into pGEX-6P-2 vector (GE Healthcare). For the mEGFP-Calaxin protein, monomerized EGFP sequence was inserted into the N-terminus of *calaxin* sequence of the plasmid. The possible motion of mEGFP tag in the mEGFP-Calaxin was suppressed by removing the linker sequence and the C-terminus region of EGFP (residue 229–238; based on *Li et al., 1997*). GST-tagged recombinant polypeptides were expressed in *E. coli* BL21 (DE3) and isolated from the cell lysate using GST-Accept beads (Nacalai Tesque) in PBS containing 0.2% Triton-X. GST tags were cleaved by HRV-3C Protease (Takara), which released the recombinant proteins from the beads. The remaining HRV-3C Protease was removed by TALON Metal Affinity Resin (Takara). In the rescue assays, 1 µM of each purified recombinant protein was added to *calaxin*[-/-] sperm axonemes in HMDEKAc buffer and incubated for 30 min on ice with 5 µM BSA as a blocking agent. Axonemes were washed once by centrifugation (5000 *g*, 3 min) and buffer exchange. Localization of mEGFP-Calaxin on *calaxin*[-/-] sperm axoneme was observed using a fluorescence microscope system (BZ-X700, Keyence). In $Ca^{2+}$ condition analysis, HMDCaKAc buffer (30 mM HEPES at pH 7.2, 5 mM $MgSO_4$, 1 mM dithiothreitol, 1 mM $CaCl_2$, and 50 mM $CH_3COOK$) was used instead of HMDEKAc buffer.

## Cryo-preparation of the zebrafish sperm axoneme

Purified sperm axonemes were diluted to desired concentrations in HMDEKAc buffer containing 0.01% Nonidet P-40. Methylated gold nanoparticles (final 1:2 dilution; CGM2K-15–25, Cytodiagnostics) were added as fiducial markers. Holey carbon grids were glow discharged to make them hydrophilic and then moisturized with a manual blot of 5 µl HMDEKAc buffer containing 0.01% Nonidet P-40 and 0.1 mg/ml BSA. Five µl of sample solution was loaded onto the grid, and then the excess liquid was removed with filter paper to make a thin film of the solution. Immediately after, the grid was plunged into liquid ethane at −180 °C for a rapid freeze. An automated plunge-freezing device (EM GP, Leica) was used to perform the blotting and freezing of the grids automatically. For $Ca^{2+}$ condition analysis, HMDCaKAc buffer was used instead of HMDEKAc buffer. Grids were stored in liquid nitrogen until the recording sessions with the electron microscope.

## Cryo-image acquisition and data processing

Cryo-images were recorded using a Krios G3i microscope (Thermo Fisher Scientific) at 300 keV, a K3 direct electron detector (Gatan) in the electron counting mode, and a Quantum-LS Energy Filter (Gatan) with a slit width of 35 eV. The magnification was set to ×15,000, which has a physical pixel size of 6.14 Å/pixel. Tilt series were acquired using the SerialEM software (*Mastronarde, 2005*) with the following settings: target defocus of 5–8 µm, angular range from −55° to 55° with 2.5° increments, image acquisition with 16–21 movie frames, and total electron dose of ~110 electrons/Å². Beam-induced motion in each movie was corrected using the MotionCor2 software (*Zheng et al., 2017*). Alignment, CTF correction, and back-projection of the tilt series images were performed using the IMOD software (*Kremer et al., 1996*) to generate the reconstructed 3D tomograms.

During the initial process of subtomographic averaging, tomograms were 4×binned to reduce the computational loads. Positions of each DMT were manually tracked using the 3dmod model tool (IMOD software), and initial particle positions were placed at about 48 nm intervals on the DMT. Alignment was performed using the PEET software (*Nicastro et al., 2006*) with volumes of 60×60 × 60 voxels in the 4×binned tomograms. The reported zebrafish DMT structure (EMD-6954; *Yamaguchi*

*et al., 2018*) was used as the initial reference. Duplicates of aligned subtomogram positions were removed at the end of the alignment process.

Based on the aligned coordinates and rotation angles of the 4×binned subtomograms, local refinement was performed in the unbinned tomograms using the PEET software again. For the refined structures of the 96 nm repeat of DMT, subtomograms were subdivided into four parts (DMT with axonemal dyneins, RS1, RS2, and RS3), and local alignment and averaging was performed individually. Obtained maps were aligned using the UCSF Chimera software (*Pettersen et al., 2004*) and combined using the EMAN2 software (*Tang et al., 2007*) as described in *Figure 3—figure supplement 1*. For refined OAD structures, four OAD repeats were extracted from the subtomograms, and then local alignment and averaging were performed. Difference maps of OAD-DC were generated by subtracting *calaxin*⁻/⁻ structure from each sample using the EMAN2 software, displayed with the same threshold of isosurface rendering. The structural classification was performed on the aligned subtomograms, using the PCA (principal component analysis) and the k-means clustering method, which are built into the PEET software (*Heumann et al., 2011*). The resolutions of the resulting structures were determined by Fourier shell correlation with a cutoff value of 0.5. The tomographic slice and averaged structures were visualized using the 3dmod program (IMOD software) and the isosurface rendering of the UCSF Chimera software, respectively.

## Immunoblot analysis

Purified *calaxin*⁻/⁻ sperm axonemes were incubated for 30 min on ice with or without 1 µM recombinant Calaxin protein in different salt concentration buffers, which were generated by adding 0, 50, 100, 150, 200, 250, or 300 mM of NaCl to HMDEKAc buffer containing 0.01% Nonidet P-40 and 5 µM BSA as a blocking agent. Axonemes were collected by centrifugation (10,000 g, 3 min) and dissolved in SDS sample buffer, followed by protein denaturation at 95 °C for 3 min. Proteins were separated by SDS-PAGE in 5–20% gradient gels (Extra PAGE One Precast Gel, Nacalai Tesque) and transferred onto polyvinylidene difluoride (PVDF) membranes (Millipore). After blocking with 5% skim milk (Nacalai Tesque) in TBST (Tris-buffered saline containing 0.1% Tween-20), membranes were incubated with primary antibodies of monoclonal anti-acetylated tubulin antibody (1:5000 dilution) and polyclonal antibodies (1:400 dilution), followed by several washes and incubation with peroxidase-conjugated secondary antibodies (1:5000 dilution). Blots were visualized by ECL Select Western Blotting Detection Reagent (GE Healthcare) and observed using a luminescent image analyzer (ImageQuant LAS4000mini, GE Healthcare). Blot signals were quantified by ImageJ/Fiji.

## Antibodies

Mouse monoclonal anti-acetylated tubulin antibody (T6793, Sigma-Aldrich) was used to visualize or quantify the zebrafish axoneme. Other primary antibodies are as follows: rabbit polyclonal anti-Dnah8 antibody (*Yamaguchi et al., 2018*), rabbit polyclonal anti-Dnah2 antibody (*Yamaguchi et al., 2018*), and guinea pig polyclonal anti-Calaxin antibody (*Sasaki et al., 2019*; referred to as anti-Efcab1 antibody). For anti-Dnah9 antibody, sequence encoding zebrafish Dnah9 (amino acid 535–1002) was subcloned into the pGEX-6P-2 plasmid vector (GE Healthcare), and recombinant polypeptide was purified from transformed *E. coli* lysate using Glutathione Sepharose 4B (GE Healthcare). Polyclonal anti-Dnah9 antibody was raised by immunization of rabbits. All polyclonal antibodies were affinity purified from serum by the antigens before use. For immunofluorescence microscopy, the following secondary antibodies were used; goat anti-rabbit IgG antibody AlexaFluor488 (A-11008, Invitrogen), goat anti-guinea pig IgG antibody AlexaFluor488 (A-11073, Invitrogen), goat anti-mouse IgG antibody AlexaFluor555 (A-21422, Invitrogen), and goat anti-guinea pig IgG antibody AlexaFluor647 (A-21450, Invitrogen). For immunoblot analysis, the following secondary antibodies were used; goat anti-mouse IgG antibody peroxidase-conjugated (A4416, Sigma-Aldrich), goat anti-rabbit IgG antibody peroxidase-conjugated (A0545, Sigma-Aldrich), and goat anti-guinea pig IgG antibody peroxidase-conjugated (A7289, Sigma-Aldrich).

## Statistics

Data with biological/technical replicates were shown using the box-and-whisker plots. The box corresponds to the first and third quartiles, the line inside the box indicates the median, and the whiskers extend to the full range of the data. In Kupffer's vesicle cilia analysis, statistical significance was tested

by Welch's t-test. In sperm motility analysis, statistical significances were tested by Tukey-Kramer multiple comparison test. p-value < 0.05 was considered to indicate a significant difference.

## Acknowledgements

We thank Dr. H Yanagisawa for his help in imaging and data processing of cryo-ET analysis.

This work was supported by JSPS KAKENHI Grant Number 16H02502 to M Kikkawa and JSPS KAKENHI Grant Number 21H04762 to M Kikkawa.

This work was also supported by Platform Project for Supporting Drug Discovery and Life Science Research (Basis for Supporting Innovative Drug Discovery and Life Science Research (BINDS)) from AMED under Grant Number JP22ama121002j001.

The authors have no competing financial interests to declare.

## Additional information

### Funding

| Funder | Grant reference number | Author |
|---|---|---|
| Japan Society for the Promotion of Science | KAKENHI Grant Number 16H02502 | Masahide Kikkawa |
| Japan Society for the Promotion of Science | KAKENHI Grant Number 21H04762 | Masahide Kikkawa |
| Japan Agency for Medical Research and Development | Grant Number JP22ama121002j001 | Masahide Kikkawa |

The funders had no role in study design, data collection and interpretation, or the decision to submit the work for publication.

### Author contributions

Hiroshi Yamaguchi, Conceptualization, Resources, Formal analysis, Investigation, Visualization, Methodology, Writing – original draft, Writing – review and editing; Motohiro Morikawa, Investigation, Writing – review and editing; Masahide Kikkawa, Conceptualization, Supervision, Funding acquisition, Methodology, Writing – original draft, Project administration, Writing – review and editing

### Author ORCIDs

Hiroshi Yamaguchi (ID) http://orcid.org/0000-0002-8722-129X
Motohiro Morikawa (ID) http://orcid.org/0000-0002-8803-9377
Masahide Kikkawa (ID) http://orcid.org/0000-0001-7656-8194

### Decision letter and Author response

Decision letter https://doi.org/10.7554/eLife.84860.sa1
Author response https://doi.org/10.7554/eLife.84860.sa2

## Additional files

### Supplementary files

• MDAR checklist

### Data availability

The Source Data files contain the numerical data and raw gel images used to generate the figures. The maps generated in this study have been deposited in EMDB under the following accession numbers: EMD-34791, EMD-34792, EMD-34793, EMD-34794, EMD-34795, EMD-34796, EMD-34797, EMD-34798, EMD-34799, EMD-34800, EMD-34801, and EMD-34802.

The following datasets were generated:

| Author(s) | Year | Dataset title | Dataset URL | Database and Identifier |
|---|---|---|---|---|
| Yamaguchi H, Kikkawa M | 2023 | Doublet microtubule of zebrafish sperm axoneme, WT | https://www.ebi.ac.uk/emdb/EMD-34791 | Electron Microscopy Data Bank, EMD-34791 |
| Yamaguchi H, Kikkawa M | 2023 | Radial spoke 1 of zebrafish sperm axoneme, WT | https://www.ebi.ac.uk/emdb/EMD-34792 | Electron Microscopy Data Bank, EMD-34792 |
| Yamaguchi H, Kikkawa M | 2023 | Radial spoke 2 of zebrafish sperm axoneme, WT | https://www.ebi.ac.uk/emdb/EMD-34793 | Electron Microscopy Data Bank, EMD-34793 |
| Yamaguchi H, Kikkawa M | 2023 | Radial spoke 3 of zebrafish sperm axoneme, WT | https://www.ebi.ac.uk/emdb/EMD-34794 | Electron Microscopy Data Bank, EMD-34794 |
| Yamaguchi H, Kikkawa M | 2023 | Doublet microtubule of zebrafish sperm axoneme, calaxin-/-, OAD+ class | https://www.ebi.ac.uk/emdb/EMD-34795 | Electron Microscopy Data Bank, EMD-34795 |
| Yamaguchi H, Kikkawa M | 2023 | Doublet microtubule of zebrafish sperm axoneme, calaxin-/-, OAD- class | https://www.ebi.ac.uk/emdb/EMD-34796 | Electron Microscopy Data Bank, EMD-34796 |
| Yamaguchi H, Kikkawa M | 2023 | Doublet microtubule of zebrafish sperm axoneme, armc4-/- | https://www.ebi.ac.uk/emdb/EMD-34797 | Electron Microscopy Data Bank, EMD-34797 |
| Yamaguchi H, Kikkawa M | 2023 | Outer arm dynein of zebrafish sperm axoneme, WT | https://www.ebi.ac.uk/emdb/EMD-34798 | Electron Microscopy Data Bank, EMD-34798 |
| Yamaguchi H, Kikkawa M | 2023 | Outer arm dynein of zebrafish sperm axoneme, WT, refined focusing on the docking complex | https://www.ebi.ac.uk/emdb/EMD-34799 | Electron Microscopy Data Bank, EMD-34799 |
| Yamaguchi H, Kikkawa M | 2023 | Outer arm dynein of zebrafish sperm axoneme, WT, 1 mM calcium condition, refined focusing on the docking complex | https://www.ebi.ac.uk/emdb/EMD-34800 | Electron Microscopy Data Bank, EMD-34800 |
| Yamaguchi H, Kikkawa M | 2023 | Outer arm dynein of zebrafish sperm axoneme, calaxin-/-, refined focusing on the docking complex | https://www.ebi.ac.uk/emdb/EMD-34801 | Electron Microscopy Data Bank, EMD-34801 |
| Yamaguchi H, Kikkawa M | 2023 | Outer arm dynein of zebrafish sperm axoneme, calaxin-/-, incubated with recombinant Calaxin protein, refined focusing on the docking complex | https://www.ebi.ac.uk/emdb/EMD-34802 | Electron Microscopy Data Bank, EMD-34802 |

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
