## [Editor Report]

In vertebrates, ciliary motility is powered by axonemal dyneins, known as OADs, tethered to doublet microtubules by a pentameric docking complex including the Armc4 and Calaxin subunits. This valuable study combines zebrafish genetics with cryo-electron tomography to convincingly show that Armc4 plays a critical role in the docking of OAD and that Calaxin stabilizes the molecular interaction. The work will be of interest to those studying the structure and function of the axoneme, and motile cilia in general.

---

## [Decision Letter]

**Decision letter after peer review:**

Thank you for submitting your article "Calaxin stabilizes the docking of outer arm dyneins onto ciliary doublet microtubule in vertebrates" for consideration by *eLife*. Your article has been reviewed by 3 peer reviewers, one of whom is a member of our Board of Reviewing Editors, and the evaluation has been overseen by Didier Stainier as the Senior Editor. The following individual involved in review of your submission has agreed to reveal their identity: Kazuo Inaba (Reviewer #2).

Essential revisions:

1) Additional immunofluorescence data are requested to demonstrate that a) Armc4 is distributed along the entire length of sperm flagella and epithelial cilia, and b) to test whether Calaxin can directly interact with β-tubulin.

2) Representative images and statistics are needed to support Figures 1F, 1G, and 3. The immunoblot in Figure 5E should be repeated to provide confidence in the findings.

3) Additional information is requested to support the cryo-ET data. These include cryo-ET images of the calaxin-/- axoneme in the presence of 1 mM EGTA, and clear indications of which regions of the flagellum are shown in Figure 5C and D. A better description of the confidence of the observations given the resolution of the cryo-ET analysis is required.

4) The purity of the recombinant Calaxin used in the rescue experiments should be shown (e.g. by SDS-PAGE analysis).

5) Some sections of the text need to be revised to improve clarity. This includes changing the description of how ODAs are tethered to the ODA-DC and the description of Calaxin as "indispensable". See the individual reviews for a full list of requested changes. Changes to the videos and the model in Figure 5F are also recommended to include more information and improve data presentation.

*Reviewer #1 (Recommendations for the authors):*

This is a focused and well-performed study that draws convincing conclusions. While there are a few areas that warrant further investigation (the hypothesized preassembly of the OAD-DC in the cytoplasm and the precise conformational changes that occur between ca^2+^ conditions), I do not believe additional experiments are necessary.

*Reviewer #2 (Recommendations for the authors):*

The model in Figure 5F for the calaxin-mediated stabilization of OAD docking is clear and accurate but I think it could include more on what the authors newly found in this study, i.e. Ca^2+^/calaxin dependent changes in the N-terminal region and possibly the distal region of CCDC151/114.

*Reviewer #3 (Recommendations for the authors):*

The authors performed IF on spermatozoa and olfactory cilia of the mutants, showing different degrees of ODA loss. Since the authors also showed defective ciliary motility of KV cilia, an immunofluorescence assay of the KV cilia would be helpful to reveal how these mutations affect ODA in a tissue-specific manner. This experiment is not essential but is a plus.

Some images and statistics need to be added. (1) Representative images of each type of heart looping are needed alongside the quantification graph in Figure 1G. (2) Particle percentage of different classes should be included in Figure 3. (3) The cilia number and p-value of calaxin-/- in Figure 1F.

I was expecting the distribution analysis of calaxin-/- OADs (Page 11, line 222-234) in the tomography section.

Page3, line 42, "OAD Defects are the most", "Defects" should be "defects".

Discussion, line 325, "Zebrafish spermatozoa have Dnah9/Dnah8 along the entire length of flagella." References should be added.

---

## [Author Response]

Essential revisions:1) Additional immunofluorescence data are requested to demonstrate that a) Armc4 is distributed along the entire length of sperm flagella and epithelial cilia, and b) to test whether Calaxin can directly interact with β-tubulin.

a) To answer this comment, we purchased a commercially available anti-ARMC4 (human) antibody. We checked the cross-reactivity of the antibody against zebrafish Armc4, but no signal was detected in our western blot analysis. Thus, we could not assess the localization of zebrafish Armc4.

However, in this manuscript, we revealed that mutation of *armc4* caused complete loss of OADs in sperm flagella (Figures 2A and 3F) and epithelial cilia (Figure 6C). These data show the general requirement of Armc4 for OADs in cilia/flagella, strongly suggesting that Armc4 is distributed along the entire length of sperm flagella and epithelial cilia.

b) In our manuscript, we wrote as follows:

(line 218-224)

“To assess the specificity of Calaxin binding, we also performed a rescue experiment with mEGFP-Calaxin (Figure 4H-I; Figure 4—figure supplement 2). *Ciona* Calaxin was reported to interact with β-tubulin (Mizuno et al., 2009), suggesting the possible binding of Calaxin along the entire length of the axoneme. However, the rescued axonemes showed partial loss of EGFP signal (Figure 4H, white arrowheads). This pattern resembled the OAD localization of *calaxin*^-/-^ in immunofluorescence microscopy, suggesting the preferential binding of Calaxin to the remaining OAD-DC. mEGFP alone showed no interaction with the axoneme (Figure 4H, asterisk).”

Therefore, our manuscript is NOT intended to support or deny the interaction between Calaxin and β-tubulin, which was reported by Mizuno et al., 2009. Instead, we focused on the interaction between Calaxin and OAD-DC, revealing that Calaxin binds to Calaxin-deficient OAD-DC (Figure 4G, H, and I). Thus, we assume this comment refers to the interaction between Calaxin and OAD-DC.

To further discuss the interaction between Calaxin and OAD-DC, we created Figure 4—figure supplement 2. We tested Calaxin’s interaction by incubating recombinant mEGFP-Calaxin with sperm axonemes of *calaxin*^-/-^, *armc4*^-/-^ (representing OAD-missing DMT), and WT (representing DMT with Calaxin and OAD). The localization of mEGFP-Calaxin was assessed by fluorescence microscopy of mEGFP signals. In *calaxin*^-/-^, mEGFP-Calaxin was bound to the limited region of the axoneme, with the partial loss of EGFP signals (Figure 4—figure supplement 2A, white arrowheads), consistent with Figure 4H. On the other hand, mEGFP-Calaxin showed no significant interaction with *armc4*^-/-^ axoneme (Figure 4—figure supplement 2B) or WT axoneme (Figure 4—figure supplement 2C). These data show the preferential binding of Calaxin to the Calaxin-deficient OAD-DC than OAD-missing DMT or WT OAD. Although Mizuno et al., 2009 reported the interaction between Calaxin and β-tubulin, our analysis could not detect the signals for such interaction, probably due to the different binding affinity of Calaxin against OAD-DC and β-tubulin.

2) Representative images and statistics are needed to support Figures 1F, 1G, and 3. The immunoblot in Figure 5E should be repeated to provide confidence in the findings.

We added the representative images (Figure 1—figure supplement 1) or numerical data (Figure 1—source data 1, 2, and 3 and Figure 3 legend) to support Figures 1F, 1G, and 3. We also created Figure 5—figure supplement 2, which shows the experimental replicate of the immunoblot analysis in Figure 5E. Specific answers are summarized with public review comments.

3) Additional information is requested to support the cryo-ET data. These include cryo-ET images of the calaxin-/- axoneme in the presence of 1 mM EGTA, and clear indications of which regions of the flagellum are shown in Figure 5C and D. A better description of the confidence of the observations given the resolution of the cryo-ET analysis is required.

We added cryo-ET images of the *calaxin*^-/-^ axoneme (1 mM EGTA condition) in Figure 7D. To clarify the region of the flagella of each cryo-ET observation, we created Figure 5—figure supplement 1. For a better description of the confidence of our cryo-ET analyses, we created Figure 7—figure supplement 1. Specific answers are summarized with public review comments.

4) The purity of the recombinant Calaxin used in the rescue experiments should be shown (e.g. by SDS-PAGE analysis).

We added Figure 4—source data 1 and 2, which show the purity of the recombinant proteins used in our study.

5) Some sections of the text need to be revised to improve clarity. This includes changing the description of how ODAs are tethered to the ODA-DC and the description of Calaxin as "indispensable". See the individual reviews for a full list of requested changes. Changes to the videos and the model in Figure 5F are also recommended to include more information and improve data presentation.

Our manuscript revealed that Calaxin is required for the stable DC-DMT docking but is unnecessary for the initial DC-DMT docking. We revised the manuscript to clarify this point, according to the reviewers' recommendations. Specific revised texts are summarized with public review comments. We also added labels in the videos to improve the data presentation.

Reviewer #2 (Recommendations for the authors):The model in Figure 5F for the calaxin-mediated stabilization of OAD docking is clear and accurate but I think it could include more on what the authors newly found in this study, i.e. Ca^2+^/calaxin dependent changes in the N-terminal region and possibly the distal region of CCDC151/114.

Thank you for pointing it out. There are at least two Calaxin functions, (a): modulating the OAD activity depending on Ca^2+^ concentrations (Mizuno et al., 2012) and (b): stabilizing the OAD docking onto DMT (this study). We speculate that Ca^2+^/Calaxin-dependent changes of CCDC151/114 are involved in Calaxin’s function (a): modulating the OAD activity depending on Ca^2+^ concentrations. On the other hand, most of our study was focused on the OAD-DMT docking, which revealed the novel Calaxin function (b): stabilizing the OAD docking onto DMT. To emphasize this point, we would like to show only the stabilizing process of OAD-DMT docking in Figure 5F.

Reviewer #3 (Recommendations for the authors):The authors performed IF on spermatozoa and olfactory cilia of the mutants, showing different degrees of ODA loss. Since the authors also showed defective ciliary motility of KV cilia, an immunofluorescence assay of the KV cilia would be helpful to reveal how these mutations affect ODA in a tissue-specific manner. This experiment is not essential but is a plus.

Immunofluorescence microscopy of KV cilia is challenging because the signal intensities varied between embryos, probably due to different efficiency of antibody penetration. Thus, we could not show the immunofluorescence microscopy data of the OADs in KV cilia.

Some images and statistics need to be added. (1) Representative images of each type of heart looping are needed alongside the quantification graph in Figure 1G. (2) Particle percentage of different classes should be included in Figure 3. (3) The cilia number and p-value of calaxin-/- in Figure 1F.

(1) We created Figure 1—figure supplement 1 to show the typical images of the mutant heart looping.

(2) We added particle percentages of each class in the Figure 3 legend:

(line 466-468)

“(D-E) DMT structures of *calaxin*^-/-^ sperm flagella. Structural classification sorted the subtomograms into two classes: (D) OAD+ class (70.7%, 6122 particles) and (E) OAD- class (29.3%, 2535 particles).”

(3) We added the numerical data of *calaxin*^-/-^ phenotypes in Figure 1—source data 1, 2, and 3. We also added p-values of *calaxin*^-/-^ data in Figure 1—source data 2. However, we did not show the p-values of *calaxin*^-/-^ data in Figure 1F, since *calaxin*^-/-^ data were derived from Sasaki et al., 2019, which collected the data of *calaxin*^-/-^ and control samples independently of this manuscript.

I was expecting the distribution analysis of calaxin-/- OADs (Page 11, line 222-234) in the tomography section.

We used 3D-reconstructed tomograms to analyze the distribution of WT and *calaxin*^-/-^ OADs in Figure 5 and Figure 5—figure supplement 1.

Page3, line 42, "OAD Defects are the most", "Defects" should be "defects".

Thank you for pointing it out. We revised the word.

Discussion, line 325, "Zebrafish spermatozoa have Dnah9/Dnah8 along the entire length of flagella." References should be added.

We created Figure 2—figure supplement 1 to show the immunofluorescence microscopy of OAD-HCs (Dnah9, OAD β-HC; Dnah8, OAD γ-HC) in zebrafish spermatozoa. We added the information on anti-Dnah9 antibody in the Materials and methods section.